# Efficient Anomaly Detection via Matrix Sketching

**Vatsal Sharan**
Stanford University*
vsharan@stanford.edu

**Parikshit Gopalan**
VMware Research
pgopalan@vmware.com

**Udi Wieder**
VMware Research
uwieder@vmware.com

## Abstract

We consider the problem of finding anomalies in high-dimensional data using popular PCA based anomaly scores. The naive algorithms for computing these scores explicitly compute the PCA of the covariance matrix which uses space quadratic in the dimensionality of the data. We give the first streaming algorithms that use space that is linear or sublinear in the dimension. We prove general results showing that *any* sketch of a matrix that satisfies a certain operator norm guarantee can be used to approximate these scores. We instantiate these results with powerful matrix sketching techniques such as Frequent Directions and random projections to derive efficient and practical algorithms for these problems, which we validate over real-world data sets. Our main technical contribution is to prove matrix perturbation inequalities for operators arising in the computation of these measures.

## 1 Introduction

Anomaly detection in high-dimensional numeric data is a ubiquitous problem in machine learning [1, 2]. A typical scenario is where we have a constant stream of measurements (say parameters regarding the health of machines in a data-center), and our goal is to detect any unusual behavior. An algorithm to detect anomalies in such high dimensional settings faces computational challenges: the dimension of the data matrix $\mathbf{A} \in \mathbb{R}^{n \times d}$ may be very large both in terms of the number of data points $n$ and their dimensionality $d$ (in the datacenter example, $d$ could be $10^6$ and $n \gg d$). The desiderata for an algorithm to be efficient in such settings are—

1. As $n$ is too large for the data to be stored in memory, the algorithm must work in a streaming fashion where it only gets a constant number of passes over the dataset.
2. As $d$ is also very large, the algorithm should ideally use memory linear or even sublinear in $d$.

In this work we focus on two popular subspace based anomaly scores: rank-$k$ leverage scores and rank-$k$ projection distance. The key idea behind subspace based anomaly scores is that real-world data often has most of its variance in a low-dimensional rank $k$ subspace, where $k$ is usually much smaller than $d$. In this section, we assume $k = O(1)$ for simplicity. These scores are based on identifying this principal $k$ subspace using Principal Component Analyis (PCA) and then computing how "normal" the projection of a point on the principal $k$ subspace looks. Rank-$k$ leverage scores compute the normality of the projection of the point *onto* the principal $k$ subspace using Mahalanobis distance, and rank-$k$ projection distance compute the $\ell_2$ distance of the point *from* the principal $k$ subspace (see Fig. 1 for an illustration). These scores have found widespread use for detection of anomalies in many applications such as finding outliers in network traffic data [3, 4, 5, 6], detecting anomalous behavior in social networks [7, 8], intrusion detection in computer security [9, 10, 11], in industrial systems for fault detection [12, 13, 14] and for monitoring data-centers [15, 16].

The standard approach to compute principal $k$ subspace based anomaly scores in a streaming setting is by computing $\mathbf{A}^T\mathbf{A}$, the $(d \times d)$ covariance matrix of the data, and then computing the top $k$

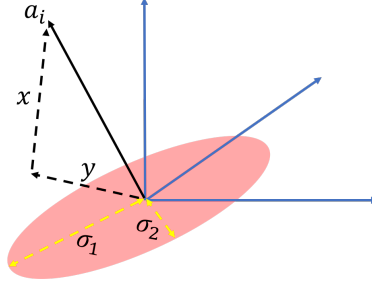

Figure 1: Illustration of subspace based anomaly scores. Here, the data lies mostly in the $k = 2$ dimensional principal subspace shaded in red. For a point $a_{(i)}$ the rank-$k$ projection distance equals $\|x\|_2$, where $x$ is the component of $a_{(i)}$ orthogonal to the principal subspace. The rank-$k$ leverage score measures the *normality* of the projection $y$ onto the principal subspace.

principal components. This takes space $O(d^2)$ and time $O(nd^2)$. The quadratic dependence on $d$ renders this approach inefficient in high dimensions. It raises the natural question of whether better algorithms exist.

## 1.1 Our Results

In this work, we answer the above question affirmatively, by giving algorithms for computing these anomaly scores that require space linear and even sublinear in $d$. Our algorithms use popular matrix sketching techniques while their analysis uses new matrix perturbation inequalities that we prove. Briefly, a sketch of a matrix produces a much smaller matrix that preserves some desirable properties of the large matrix (formally, it is close in some suitable norm). Sketching techniques have found numerous applications to numerical linear algebra. Several efficient sketching algorithms are known in the streaming setting [17].

**Pointwise guarantees with linear space:** We show that any sketch $\tilde{\mathbf{A}}$ of $\mathbf{A}$ with the property that $\|\mathbf{A}^T\mathbf{A} - \tilde{\mathbf{A}}^T\tilde{\mathbf{A}}\|$ is small, can be used to additively approximate the rank-$k$ leverage scores and rank-$k$ projection distances for each row. By instantiating this with suitable sketches such as the Frequent Directions sketch [18], row-sampling [19] or a random projection of the columns of the input, we get a streaming algorithm that uses $O(d)$ memory and $O(nd)$ time.

**A matching lower bound:** Can we get such an additive approximation using memory only $o(d)$?[2] The answer is no, we show a lower bound saying that any algorithm that computes such an approximation to the rank-$k$ leverage scores or the rank-$k$ projection distances for all the rows of a matrix must use $\Omega(d)$ working space, using techniques from communication complexity. Hence our algorithm has near-optimal dependence on $d$ for the task of approximating the outlier scores for every data point.

**Average-case guarantees with logarithmic space:** Perhaps surprisingly, we show that it is actually possible to circumvent the lower bound by relaxing the requirement that the outlier scores be preserved for each and every point to only preserving the outlier scores on average. For this we require sketches where $\|\mathbf{A}\mathbf{A}^T - \tilde{\mathbf{A}}\tilde{\mathbf{A}}^T\|$ is small: this can be achieved via random projection of the rows of the input matrix or column subsampling [19]. Using any such sketch, we give a streaming algorithm that can preserve the outlier scores for the rows up to small additive error on average, and hence preserve most outliers. The space required by this algorithm is only $\text{poly}(k)\log(d)$, and hence we get significant space savings in this setting (recall that we assume $k = O(1)$).

**Technical contributions.** A sketch of a matrix $\mathbf{A}$ is a significantly smaller matrix $\tilde{\mathbf{A}}$ which approximates it well in some norm, say for instance $\|\mathbf{A}^T\mathbf{A} - \tilde{\mathbf{A}}^T\tilde{\mathbf{A}}\|$ is small. We can think of such a sketch as a noisy approximation of the true matrix. In order to use such sketches for anomaly

detection, we need to understand how the noise affects the anomaly scores of the rows of the matrix. Matrix perturbation theory studies the effect of adding noise to the spectral properties of a matrix, which makes it the natural tool for us. The basic results here include Weyl's inequality [20] and Wedin's theorem [21], which respectively give such bounds for eigenvalues and eigenvectors. We use these results to derive perturbation bounds on more complex projection operators that arise while computing outlier scores, these operators involve projecting onto the top-$k$ principal subspace, and rescaling each co-ordinate by some function of the corresponding singular values. We believe these results could be of independent interest.

**Experimental results.**   Our results have a parameter $\ell$ that controls the size and the accuracy of the sketch. While our theorems imply that $\ell$ can be chosen independent of $d$, they depend polynomially on $k$, the desired accuracy and other parameters, and are probably pessimistic. We validate both our algorithms on real world data. In our experiments, we found that choosing $\ell$ to be a small multiple of $k$ was sufficient to get good results. Our results show that one can get outcomes comparable to running full-blown SVD using sketches which are significantly smaller in memory footprint, faster to compute and easy to implement (literally a few lines of Python code).

This contributes to a line of work that aims to make SVD/PCA scale to massive datasets [22]. We give simple and practical algorithms for anomaly score computation, that give SVD-like guarantees at a significantly lower cost in terms of memory, computation and communication.

## 2   Notation and Setup

Given a matrix $\mathbf{A} \in \mathbb{R}^{n \times d}$, we let $a_{(i)} \in \mathbb{R}^d$ denote its $i^{th}$ row and $a^{(i)} \in \mathbb{R}^n$ denote its $i^{th}$ column. Let $\mathbf{U\Sigma V}^T$ be the SVD of $\mathbf{A}$ where $\mathbf{\Sigma} = \mathrm{diag}(\sigma_1, \ldots, \sigma_d)$, for $\sigma_1 \geq \cdots \geq \sigma_d > 0$. Let $\kappa_k$ be the condition number of the top $k$ subspace of $\mathbf{A}$, defined as $\kappa_k = \sigma_1^2/\sigma_k^2$. We consider all vectors as column vectors (that includes $a_{(i)}$). We denote by $\|\mathbf{A}\|_{\mathrm{F}}$ the Frobenius norm of the matrix, and by $\|\mathbf{A}\|$ the operator norm (which is equal to the largest singular value). Subspace based measures of anomalies have their origins in a classical metric in statistics known as Mahalanobis distance, denoted by $L(i)$ and defined as,

$$L(i) = \sum_{j=1}^{d} (a_{(i)}^T v^{(j)})^2/\sigma_j^2, \tag{1}$$

where $a_{(i)}$ and $v^{(i)}$ are the $i^{th}$ row of $\mathbf{A}$ and $i^{th}$ column of $\mathbf{V}$ respectively. $L(i)$ is also known as the *leverage score* [23, 24]. If the data is drawn from a multivariate Gaussian distribution, then $L(i)$ is proportional to the negative log likelihood of the data point, and hence is the right anomaly metric in this case. Note that the higher leverage scores correspond to outliers in the data.

However, $L(i)$ depends on the entire spectrum of singular values and is highly sensitive to smaller singular values, whereas real world data sets often have most of their signal in the top singular values. Therefore the above sum is often limited to only the $k$ largest singular values (for some appropriately chosen $k \ll d$) [1, 25]. This measure is called the rank $k$ leverage score $L^k(i)$, where

$$L^k(i) = \sum_{j=1}^{k} (a_{(i)}^T v^{(j)})^2/\sigma_j^2.$$

The rank $k$ leverage score is concerned with the mass which lies within the principal space, but to catch anomalies that are far from the principal subspace a second measure of anomaly is the rank $k$ *projection distance* $T^k(i)$, which is simply the distance of the data point $a_{(i)}$ to the rank $k$ principal subspace—

$$T^k(i) = \sum_{j=k+1}^{d} (a_{(i)}^T v^{(j)})^2.$$

**Assumptions.**   We now discuss assumptions needed for our anomaly scores to be meaningful.

*(1) Separation assumption.* If there is degeneracy in the spectrum of the matrix, namely that $\sigma_k^2 = \sigma_{k+1}^2$ then the $k$-dimensional principal subspace is not unique, and then the quantities $L^k$ and $T^k$ are not well defined, since their value will depend on the choice of principal subspace. This

suggests that we are using the *wrong* value of $k$, since the choice of $k$ ought to be such that the directions orthogonal to the principal subspace have markedly less variance than those in the principal subspace. Hence we require that $k$ is such that there is a gap in the spectrum at $k$.

**Assumption 1.** *We define a matrix* $\mathbf{A}$ *as being* $(k, \Delta)$*-separated if* $\sigma_k^2 - \sigma_{k+1}^2 \geq \Delta\sigma_1^2$. *Our results assume that the data are* $(k, \Delta)$*-separated for* $\Delta > 0$.

This assumptions manifests itself as an inverse polynomial dependence on $\Delta$ in our bounds. This dependence is probably pessimistic: in our experiments, we have found our algorithms do well on datasets which are not degenerate, but where the separation $\Delta$ is not particularly large.

*(2) Approximate low-rank assumption.* We assume that the top-$k$ principal subspace captures a constant fraction (at least $0.1$) of the total variance in the data, formalized as follows.

**Assumption 2.** *We assume the matrix* $\mathbf{A}$ *is approximately rank-k, i.e.,* $\sum_{i=1}^{k} \sigma_i^2 \geq (1/10) \sum_{i=1}^{d} \sigma_i^2$.

From a technical standpoint, this assumption is not strictly needed: if Assumption 2 is not true, our results still hold, but in this case they depend on the stable rank $\mathrm{sr}(\mathbf{A})$ of $\mathbf{A}$, defined as $\mathrm{sr}(\mathbf{A}) = \sum_{i=1}^{d} \sigma_i^2 / \sigma_1^2$ (we state these general forms of our results in the appendix).

From a practical standpoint though, this assumption captures the setting where the scores $L^k$ and $T^k$, and our guarantees are most meaningful. Indeed, our experiments suggest that our algorithms work best on data sets where relatively few principal components explain most of the variance.

**Setup.** We work in the row-streaming model, where rows appear one after the other in time. Note that the leverage score of a row depends on the entire matrix, and hence computing the anomaly scores in the streaming model requires care, since if the rows are seen in streaming order, when row $i$ arrives we cannot compute its leverage score without seeing the rest of the input. Indeed, 1-pass algorithms are not possible (unless they output the entire matrix of scores at the end of the pass, which clearly requires a lot of memory). Hence we will aim for 2-pass algorithms.

Note that there is a simple 2-pass algorithm which uses $O(d^2)$ memory to compute the covariance matrix in one pass, then computes its SVD, and using this computes $L^k(i)$ and $T^k(i)$ in a second pass using memory $O(dk)$. This requires $O(d^2)$ memory and $O(nd^2)$ time, and our goal would be to reduce this to linear or sublinear in $d$.

Another reasonable way to define leverage scores and projection distances in the streaming model is to define them with respect to only the input seen so far. We refer to this as the online scenario, and refer to these scores as the online scores. Our result for sketches which preserve row spaces also hold in this online scenario. We defer more discussion of this online scenario to the appendix, and focus here only on the scores defined with respect to the entire matrix for simplicity.

## 3 Guarantees for anomaly detection via sketching

Our main results say that given $\mu > 0$ and a $(k, \Delta)$-separated matrix $\mathbf{A} \in \mathbb{R}^{n \times d}$ with top singular value $\sigma_1$, any sketch $\tilde{\mathbf{A}} \in \mathbb{R}^{\ell \times d}$ satisfying

$$\|\mathbf{A}^T\mathbf{A} - \tilde{\mathbf{A}}^T\tilde{\mathbf{A}}\| \leq \mu\sigma_1^2, \tag{2}$$

or a sketch $\tilde{\mathbf{A}} \in \mathbb{R}^{n \times \ell}$ satisfying

$$\|\mathbf{A}\mathbf{A}^T - \tilde{\mathbf{A}}\tilde{\mathbf{A}}^T\| \leq \mu\sigma_1^2, \tag{3}$$

can be used to approximate rank $k$ leverage scores and the projection distance from the principal $k$-dimensional subspace. The quality of the approximation depends on $\mu$, the separation $\Delta$, $k$ and the condition number $\kappa_k$ of the top $k$ subspace.[3] In order for the sketches to be useful, we also need them to be efficiently computable in a streaming fashion. We show how to use such sketches to design efficient algorithms for finding anomalies in a streaming fashion using small space and with fast running time. The actual guarantees (and the proofs) for the two cases are different and incomparable. This is to be expected as the sketch guarantees are very different in the two cases: Equation (2) can be viewed as an approximation to the covariance matrix of the row vectors, whereas Equation (3)

gives an approximation for the covariance matrix of the column vectors. Since the corresponding sketches can be viewed as preserving the row/column space of $\mathbf{A}$ respectively, we will refer to them as row/column space approximations.

**Pointwise guarantees from row space approximations.** Sketches which satisfy Equation (2) can be computed in the row streaming model using random projections of the columns, subsampling the rows of the matrix proportional to their squared lengths [19] or deterministically by using the Frequent Directions algorithm [26]. Our streaming algorithm is stated as Algorithm 1, and is very simple. In Algorithm 1, any other sketch such as subsampling the rows of the matrix or using a random projection can also be used instead of Frequent Directions.

---

**Algorithm 1:** Algorithm to approximate anomaly scores using Frequent Directions

**Input**: Choice of $k$, sketch size $\ell$ for Frequent Directions [26]
**First Pass:**
    Use Frequent Directions to compute a sketch $\tilde{\mathbf{A}} \in \mathbb{R}^{\ell \times d}$
**SVD:**
    Compute the top $k$ right singular vectors of $\tilde{\mathbf{A}}^T \tilde{\mathbf{A}}$
**Second Pass:** *As each row $a_{(i)}$ streams in,*
    Use estimated right singular vectors to compute leverage scores and projection distances

---

We state our results here, see Section B for precise statements and general results for any sketches which satisfy the guarantee in Eq. (2). All our proofs are deferred to the appendix in the supplementary material.

**Theorem 1.** *Assume that $\mathbf{A}$ is $(k, \Delta)$-separated. There exists $\ell = k^2 \cdot \mathrm{poly}(\varepsilon^{-1}, \kappa_k, \Delta)$, such that the above algorithm computes estimates $\tilde{T}^k(i)$ and $\tilde{L}^k(i)$ where*

$$|T^k(i) - \tilde{T}^k(i)| \leq \varepsilon \|a_{(i)}\|_2^2,$$

$$|L^k(i) - \tilde{L}^k(i)| \leq \varepsilon k \frac{\|a_{(i)}\|_2^2}{\|\mathbf{A}\|_F^2}.$$

*The algorithm uses memory $O(d\ell)$ and has running time $O(nd\ell)$.*

The key is that while $\ell$ depends on $k$ and other parameters, it is independent of $d$. In the setting where all these parameters are constants independent of $d$, our memory requirement is $O(d)$, improving on the trivial $O(d^2)$ bound.

Our approximations are additive rather than multiplicative. But for anomaly detection, the candidate anomalies are ones where $L^k(i)$ or $T^k(i)$ is large, and in this regime, we argue below that our additive bounds also translate to good multiplicative approximations. The additive error in computing $L^k(i)$ is about $\varepsilon k/n$ when all the rows have roughly equal norm. Note that the average rank-$k$ leverage score of all the rows of any matrix with $n$ rows is $k/n$, hence a reasonable threshold on $L^k(i)$ to regard a point as an anomaly is when $L^k(i) \gg k/n$, so the guarantee for $L^k(i)$ in Theorem 1 preserves anomaly scores up to a small multiplicative error for candidate anomalies, and ensures that points which were not anomalies before are not mistakenly classified as anomalies. For $T^k(i)$, the additive error for row $a_{(i)}$ is $\varepsilon \|a_{(i)}\|_2^2$. Again, for points that are anomalies, $T^k(i)$ is a constant fraction of $\|a_{(i)}\|_2^2$, so this guarantee is meaningful.

Next we show that substantial savings are unlikely for any algorithm with strong pointwise guarantees: there is an $\Omega(d)$ lower bound for any approximation that lets you distinguish $L^k(i) = 1$ from $L^k(i) = \varepsilon$ for any constant $\varepsilon$.

**Theorem 2.** *Any streaming algorithm which takes a constant number of passes over the data and can compute a $0.1$ error additive approximation to the rank-$k$ leverage scores or the rank-$k$ projection distances for all the rows of a matrix must use $\Omega(d)$ working space.*

**Average-case guarantees from columns space approximations.** We derive smaller space algorithms, albeit with weaker guarantees using sketches that give columns space approximations that satisfy Equation (3). Even though the sketch gives column space approximations our goal is still to

compute the row anomaly scores, so it not just a matter of working with the transpose. Many sketches are known which approximate $\mathbf{A}\mathbf{A}^T$ and satisfy Equation (3), for instance, a low-dimensional projection by a random matrix $\mathbf{R} \in \mathbb{R}^{d \times \ell}$ (e.g., each entry of $\mathbf{R}$ could be a scaled i.i.d. uniform $\{\pm 1\}$ random variable) satisfies Equation (3) for $\ell = O(k/\mu^2)$ [27].

On first glance it is unclear how such a sketch should be useful: the matrix $\tilde{\mathbf{A}}\tilde{\mathbf{A}}^T$ is an $n \times n$ matrix, and since $n \gg d$ this matrix is too expensive to store. Our streaming algorithm avoids this problem by only computing $\tilde{\mathbf{A}}^T\tilde{\mathbf{A}}$, which is an $\ell \times \ell$ matrix, and the larger matrix $\tilde{\mathbf{A}}\tilde{\mathbf{A}}^T$ is only used for the analysis. Instantiated with the sketch above, the resulting algorithm is simple to describe (although the analysis is subtle): we pick a random matrix in $\mathbf{R} \in \mathbb{R}^{d \times \ell}$ as above and return the anomaly scores for the sketch $\tilde{\mathbf{A}} = \mathbf{A}\mathbf{R}$ instead. Doing this in a streaming fashion using even the naive algorithm requires computing the small covariance matrix $\tilde{\mathbf{A}}^T\tilde{\mathbf{A}}$, which is only $O(\ell^2)$ space.

But notice that we have not accounted for the space needed to store the $(d \times \ell)$ matrix $\mathbf{R}$. This is a subtle (but mainly theoretical) concern, which can be addressed by using powerful results from the theory of pseudorandomness [28]. Constructions of pseudorandom Johnson-Lindenstrauss matrices [29, 30] imply that the matrix $\mathbf{R}$ can be pseudorandom, meaning that it has a succinct description using only $O(\log(d))$ bits, from which each entry can be efficiently computed on the fly.

---

**Algorithm 2:** Algorithm to approximate anomaly scores using random projection

**Input**: Choice of $k$, random projection matrix $\mathbf{R} \in \mathbb{R}^{d \times \ell}$
**Initialization**
$\quad$ | $\quad$ Set covariance $\tilde{\mathbf{A}}^T\tilde{\mathbf{A}} \leftarrow 0$
**First Pass:** *As each row $a_{(i)}$ streams in,*
$\quad$ | $\quad$ Project by $\mathbf{R}$ to get $\mathbf{R}^T a_{(i)}$
$\quad$ | $\quad$ Update covariance $\tilde{\mathbf{A}}^T\tilde{\mathbf{A}} \leftarrow \tilde{\mathbf{A}}^T\tilde{\mathbf{A}} + (\mathbf{R}^T a_{(i)})(\mathbf{R}^T a_{(i)})^T$
**SVD:**
$\quad$ | $\quad$ Compute the top $k$ right singular vectors of $\tilde{\mathbf{A}}^T\tilde{\mathbf{A}}$
**Second Pass:** *As each row $a_{(i)}$ streams in,*
$\quad$ | $\quad$ Project by $\mathbf{R}$ to get $\mathbf{R}^T a_{(i)}$
$\quad$ | $\quad$ For each projected row, use the estimated right singular vectors to compute the leverage scores and projection distances

---

**Theorem 3.** *For $\varepsilon$ sufficiently small, there exists $\ell = k^3 \cdot \mathrm{poly}(\varepsilon^{-1}, \Delta)$ such that the algorithm above produces estimates $\tilde{L}^k(i)$ and $\tilde{T}^k(i)$ in the second pass, such that with high probabilty,*

$$\sum_{i=1}^n |T^k(i) - \tilde{T}^k(i)| \leq \varepsilon \|\mathbf{A}\|_{\mathrm{F}}^2,$$

$$\sum_{i=1}^n |L^k(i) - \tilde{L}^k(i)| \leq \varepsilon \sum_{i=1}^n L^k(i).$$

*The algorithm uses space $O(\ell^2 + \log(d)\log(k))$ and has running time $O(nd\ell)$.*

This gives an average case guarantee. We note that Theorem 3 shows a new property of random projections—that on average they can preserve leverage scores and distances from the principal subspace, with the projection dimension $\ell$ being only $\mathrm{poly}(k, \varepsilon^{-1}, \Delta)$, independent of both $n$ and $d$.

We can obtain similar guarantees as in Theorem 3 for other sketches which preserve the column space, such as sampling the columns proportional to their squared lengths [19, 31], at the price of one extra pass. Again the resulting algorithm is very simple: it maintains a carefully chosen $\ell \times \ell$ submatrix of the full $d \times d$ covariance matrix $\mathbf{A}^T\mathbf{A}$ where $\ell = O(k^3)$. We state the full algorithm in Section C.3.

## 4 Experimental evaluation

The aim of our experiments is to test whether our algorithms give comparable results to exact anomaly score computation based on full SVD. So in our experiments, we take the results of SVD as the

ground truth and see how close our algorithms get to it. In particular, the goal is to determine how large the parameter $\ell$ that determines the size of the sketch needs to be to get close to the exact scores. Our results suggest that for high dimensional data sets, it is possible to get good approximations to the exact anomaly scores even for fairly small values of $\ell$ (a small multiple of $k$), hence our worst-case theoretical bounds (which involve polynomials in $k$ and other parameters) are on the pessimistic side.

**Datasets:** We ran experiments on three publicly available datasets: p53 mutants [32], Dorothea [33] and RCV1 [34], all of which are available from the UCI Machine Learning Repository, and are high dimensional ($d > 5000$). The original RCV1 dataset contains 804414 rows, we took every tenth element from it. The sizes of the datasets are listed in Table 1.

**Ground Truth:** To establish the ground truth, there are two parameters: the dimension $k$ (typically between 10 and 125) and a threshold $\eta$ (typically between 0.01 and 0.1). We compute the anomaly scores for this $k$ using a full SVD, and then label the $\eta$ fraction of points with the highest anomaly scores to be outliers. $k$ is chosen by examining the explained variance of the datatset as a function of $k$, and $\eta$ by examining the histogram of the anomaly score.

**Our Algorithms:** We run Algorithm 1 using random column projections in place of Frequent Directions.[4] The relevant parameter here is the projection dimension $\ell$, which results in a sketch matrix of size $d \times \ell$. We run Algorithm 2 with random row projections. If the projection dimension is $\ell$, the resulting sketch size is $O(\ell^2)$ for the covariance matrix. For a given $\ell$, the time complexity of both algorithms is similar, however the size of the sketches are very different: $O(d\ell)$ versus $O(\ell^2)$.

**Measuring accuracy:** We ran experiments with a range of $\ell$s, in the range $(2k, 20k)$ for each dataset (hence the curves have different start/end points). The algorithm is given just the points (without labels or $\eta$) and computes anomaly scores for them. We then declare the points with the top $\eta'$ fraction of scores to be anomalies, and then compute the $F_1$ score (defined as the harmonic mean of the precision and the recall). We choose the value of $\eta'$ which maximizes the $F_1$ score. This measures how well the proposed algorithms can approximate the exact outlier scores. Note that in order to get both good precision and recall, $\eta'$ cannot be too far from $\eta$. We report the average $F_1$ score over 5 runs.

For each dataset, we run both algorithms, approximate both the leverage and projection scores, and try three different values of $k$. For each of these settings, we run over roughly 10 values for $\ell$. The results are plotted in Figs. 2, 3 and 4. Here are some takeaways from our experiments:

- Taking $\ell = Ck$ with a fairly small $C \approx 10$ suffices to get F1 scores $> 0.75$ in most settings.

- Algorithm 1 generally outperforms Algorithm 2 for a given value of $\ell$. This should not be too surprising given that it uses much more memory, and is known to give pointwise rather than average case guarantees. However, Algorithm 2 does surprisingly well for an algorithm whose memory footprint is essentially independent of the input dimension $d$.

- The separation assumption (Assumption (1)) does hold to the extent that the spectrum is not degenerate, but not with a large gap. The algorithms seem fairly robust to this.

- The approximate low-rank assumption (Assumption (2)) seems to be important in practice. Our best results are for the p53 data set, where the top 10 components explain $87\%$ of the total variance. The worst results are for the RCV1 data set, where the top 100 and 200 components explain only $15\%$ and $25\%$ of the total variance respectively.

**Performance.** While the main focus of this work is on the streaming model and memory consumption, our algorithms offer considerable speedups even in the offline/batch setting. Our timing experiments were run using Python/Jupyter notebook on a linux VM with 8 cores and 32 Gb of RAM, the times reported are total CPU times in seconds as measured by the % time function, and are reported in Table 1. We focus on computing projection distances using SVD (the baseline), Random Column Projection (Algorithm 1) and Random Row Projection (Algorithm 2). All SVD computations use the `randomized_svd` function from `scikit.learn`. The baseline computes only the top $k$ singular values and vectors (not the entire SVD). The results show consistent speedups between $2\times$ and $6\times$. Which algorithm is faster depends on which dimension of the input matrix is larger.

Table 1: Running times for computing rank-$k$ projection distance. Speedups between $2\times$ and $6\times$.

| Dataset | Size ($n \times d$) | $k$ | $\ell$ | SVD | Column Projection | Row Projection |
|---|---|---|---|---|---|---|
| p53 mutants | $16772 \times 5409$ | 20 | 200 | 29.2s | 6.88s | 7.5s |
| Dorothea | $1950 \times 100000$ | 20 | 200 | 17.7s | 9.91s | 2.58s |
| RCV1 | $80442 \times 47236$ | 50 | 500 | 39.6s | 17.5s | 20.8s |

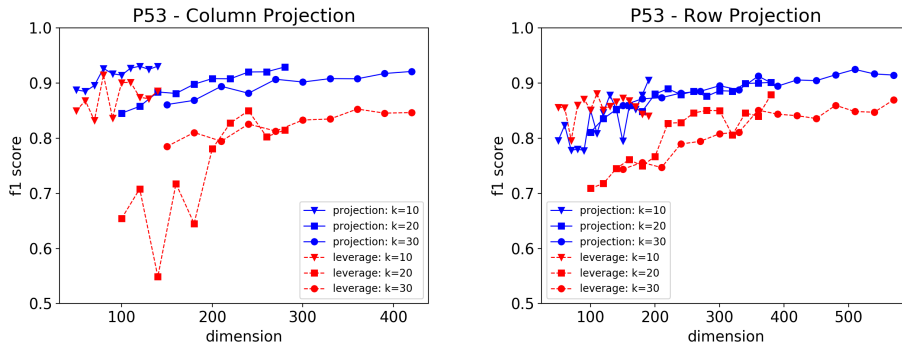

Figure 2: Results for P53 Mutants. We get $F_1$ score $> 0.8$ with $> 10\times$ space savings.

## 5 Related work

In most anomaly detection settings, labels are hard to come by and unsupervised learning methods are preferred: the algorithm needs to learn what the bulk of the data looks like and then detect any deviations from this. Subspace based scores are well-suited to this, but various other anomaly scores have also been proposed such as those based on approximating the density of the data [36, 37] and attribute-wise analysis [38], we refer to surveys on anomaly detection for an overview [1, 2].

Leverage scores have found numerous applications in numerical linear algebra, and hence there has been significant interest in improving the time complexity of computing them. For the problem of approximating the (full) leverage scores ($L(i)$ in Eq. (1), note that we are concerned with the rank-$k$ leverage scores $L^k(i)$), Clarkson and Woodruff [39] and Drineas et al. [40] use sparse subspace embeddings and Fast Johnson Lindenstrauss Transforms (FJLT [41]) to compute the leverage scores using $O(nd)$ time instead of the $O(nd^2)$ time required by the baseline—but these still need $O(d^2)$ memory. With respect to projection distance, the closest work to ours is Huang and Kasiviswanathan [42] which uses Frequent Directions to approximate projection distances in $O(kd)$ space. In contrast to these approaches, our results hold both for rank-$k$ leverage scores and projection distances, for any matrix sketching algorithm—not just FJLT or Frequent Directions—and our space requirement can be as small as $\log(d)$ for average case guarantees. However, Clarkson and Woodruff [39] and Drineas et al. [40] give multiplicative guarantees for approximating leverage scores while our guarantees for rank-$k$ leverage scores are additive, but are nevertheless sufficient for the task of detecting anomalies.

## 6 Conclusion

We show that techniques from sketching can be used to derive simple and practical algorithms for computing subspace-based anomaly scores which provably approximate the true scores at a significantly lower cost in terms of time and memory. A promising direction of future work is to use them in real-world high-dimensional anomaly detection tasks.

## Acknowledgments

The authors thank David Woodruff for suggestions on using communication complexity tools to show lower bounds on memory usage for approximating anomaly scores and Weihao Kong for several useful discussions on estimating singular values and vectors using random projections. We also thank Steve Mussmann, Neha Gupta, Yair Carmon and the anonymous reviewers for detailed feedback on

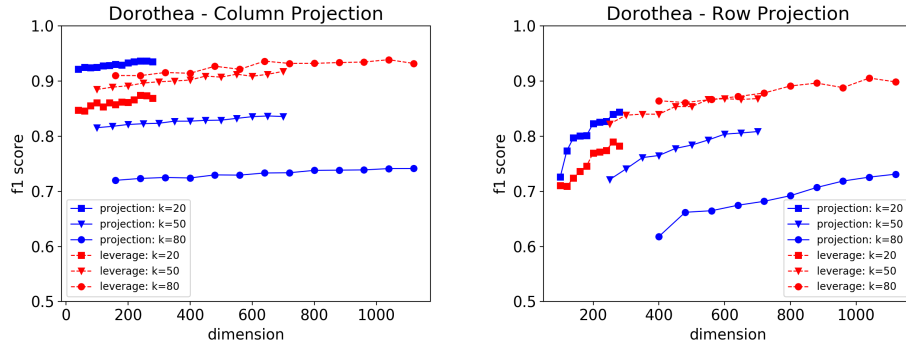

Figure 3: Results for the Dorothea dataset. Column projections give more accurate approximations, but they use more space.

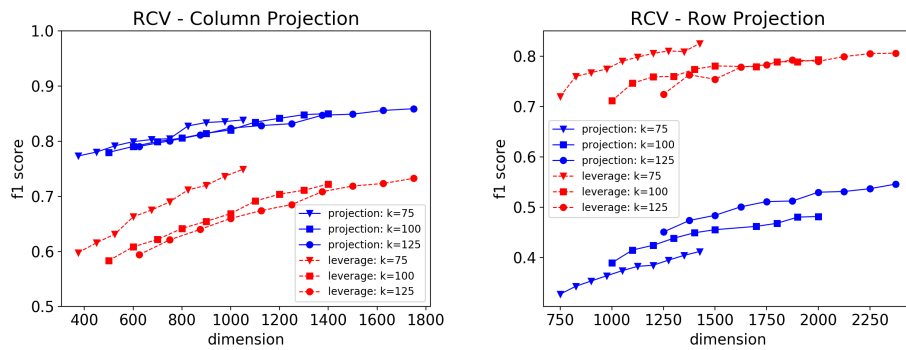

Figure 4: Results for the RCV1 dataset. Our results here are worse than for the other datasets, we hypothesize this is due to this data having less pronounced low-rank structure.

initial versions of the paper. VS's contribution was partially supported by NSF award 1813049, and ONR award N00014-18-1-2295.

## Footnotes

*Part of the work was done while the author was an intern at VMware Research.

[2]Note that even though each row is $d$ dimensional an algorithm need not store the entire row in memory, and could instead perform computations as each coordinate of the row streams in.

[3]The dependence on $\kappa_k$ only appears for showing guarantees for rank-$k$ leverage scores $L^k$ in Theorem 1.

[4]Since the existing implementation of Frequent Directions [35] does not seem to handle sparse matrices.

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
