[Supplementary Material]

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

# A  Matrix Perturbation Bounds

In this section, we will establish projection bounds for various operators needed for computing outlier scores. We first set up some notation and state some results we need.

## A.1  Preliminaries

We work with the following setup throughout this section. Let $\mathbf{A} \in \mathbb{R}^{n \times d} = \mathbf{U}\mathbf{\Sigma}\mathbf{V}^T$ where $\mathbf{\Sigma} = (\sigma_1, \dots, \sigma_d)$. Assume that $\mathbf{A}$ is $(k, \Delta)$-separated as in Assumption 1 from 2. We use $\mathrm{sr}(\mathbf{A}) = \|\mathbf{A}\|_{\mathrm{F}}^2 / \sigma_1^2$ to denote the stable rank of $\mathbf{A}$, and $\kappa_k = \sigma_1^2 / \sigma_k^2$ for the condition number of $\mathbf{A}_k$.

Let $\tilde{\mathbf{A}} \in \mathbb{R}^{n \times \ell}$ be a sketch/noisy version of $\mathbf{A}$ satisfying

$$\|\mathbf{A}\mathbf{A}^T - \tilde{\mathbf{A}}\tilde{\mathbf{A}}^T\| \leq \mu\sigma_1^2. \tag{4}$$

and let $\tilde{\mathbf{A}} = \tilde{\mathbf{U}}\tilde{\mathbf{\Sigma}}\tilde{\mathbf{V}}^T$ denote its SVD. While we did not assume $\tilde{\mathbf{A}}$ is $(k, O(\Delta))$-separated, it will follow from Weyl's inequality for $\mu$ sufficiently small compared to $\Delta$. It helps to think of $\Delta$ as property of the input $\mathbf{A}$, and $\mu$ as an accuracy parameter that we control.

In this section we prove perturbation bounds for the following three operators derived from $\tilde{\mathbf{A}}$, showing their closeness to those derived from $\mathbf{A}$:

1. $\mathbf{U}_k\mathbf{U}_k^T$: projects onto the principal $k$-dimensional column subspace (Lemma 3).

2. $\mathbf{U}_k\mathbf{\Sigma}_k^2\mathbf{U}_k^T$: projects onto the principal $k$-dimensional column subspace, and scale coordinate $i$ by $\sigma_i$ (Theorem 4).

3. $\mathbf{U}_k\mathbf{\Sigma}_k^{-2}\mathbf{U}_k^T$: projects onto the principal $k$-dimensional column subspace, and scale coordinate $i$ by $1/\sigma_i$ (Theorem 6).

To do so, we will extensively use two classical results about matrix perturbation: Weyl's inequality (c.f. Horn and Johnson [20] Theorem 3.3.16) and Wedin's theorem [21], which respectively quantify how the eigenvalues and eigenvectors of a matrix change under perturbations.

**Lemma 1. (Weyl's inequality)** *Let* $\mathbf{C}, \mathbf{D} = \mathbf{C} + \mathbf{N} \in \mathbb{R}^{n \times d}$. *Then for all* $i \leq \min(n, d)$,

$$|\sigma_i(\mathbf{C}) - \sigma_i(\mathbf{D})| \leq \|\mathbf{N}\|.$$

Wedin's theorem requires a sufficiently large separation in the eigenvalue spectrum for the bound to be meaningful.

**Lemma 2. (Wedin's theorem)** *Let* $\mathbf{C}, \mathbf{D} = \mathbf{C} + \mathbf{N} \in \mathbb{R}^{n \times d}$. *Let* $\mathbf{P}_C$ *and* $\mathbf{P}_D$ *respectively denote the projection matrix onto the space spanned by the top* $k$ *singular vectors of* $\mathbf{C}$ *and* $\mathbf{D}$. *Then,*

$$\|\mathbf{P}_D - \mathbf{P}_C\mathbf{P}_D\| \leq \frac{\|\mathbf{N}\|}{\sigma_k(\mathbf{C}) - \sigma_{k+1}(\mathbf{C}) - \|\mathbf{N}\|}.$$

## A.2  Matrix Perturbation Bounds

We now use these results to derive the perturbation bounds enumerated above. The first bound is a direct consequence of Wedin's theorem.

**Lemma 3.** *If* $\mathbf{A} = \mathbf{U}\mathbf{\Sigma}\mathbf{V}^T$ *is* $(k, \Delta)$-*separated,* $\tilde{\mathbf{A}} = \tilde{\mathbf{U}}\tilde{\mathbf{\Sigma}}\tilde{\mathbf{V}}^T$ *satisfies* (4) *with* $\mu \leq \Delta/6$, *then*

$$\|\mathbf{U}_k\mathbf{U}_k^T - \tilde{\mathbf{U}}_k\tilde{\mathbf{U}}_k^T\| \leq 2\sqrt{\frac{\mu}{\Delta}}.$$

**Proof.** Let $\mathbf{P} = \mathbf{U}_k\mathbf{U}_k^T$ and $\tilde{\mathbf{P}} = \tilde{\mathbf{U}}_k\tilde{\mathbf{U}}_k^T$. Since we have $\mathbf{A}\mathbf{A}^T = \mathbf{U}\mathbf{\Sigma}^2\mathbf{U}^T$, $\mathbf{P}$ ($\tilde{\mathbf{P}}$) is the projection operator onto the column space of $\mathbf{A}_k$ (and similarly for $\tilde{\mathbf{P}}$ and $\tilde{\mathbf{A}}_k$). Since $\mathbf{P}^T = \mathbf{P}$ and $\mathbf{P}\mathbf{P} = \mathbf{P}$ for any orthogonal projection matrix, we can write,

$$\|\mathbf{P} - \tilde{\mathbf{P}}\|^2 = \|(\mathbf{P} - \tilde{\mathbf{P}})^2\| = \|\mathbf{P}\mathbf{P} - \mathbf{P}\tilde{\mathbf{P}} - \tilde{\mathbf{P}}\mathbf{P} + \tilde{\mathbf{P}}\tilde{\mathbf{P}}\| = \|\mathbf{P} - \mathbf{P}\tilde{\mathbf{P}} + \tilde{\mathbf{P}} - \tilde{\mathbf{P}}\mathbf{P}\|$$
$$\leq \|\tilde{\mathbf{P}} - \mathbf{P}\tilde{\mathbf{P}}\| + \|\mathbf{P} - \tilde{\mathbf{P}}\mathbf{P}\|. \tag{5}$$

Since $\mathbf{A}$ is $(k, \Delta)$-separated,

$$\sigma_k(\mathbf{A}\mathbf{A}^t) - \sigma_{k+1}(\mathbf{A}\mathbf{A}^t) = \sigma_k^2 - \sigma_{k+1}(\mathbf{A})^2 = \Delta. \tag{6}$$

So applying Wedin's theorem to $\mathbf{A}\mathbf{A}^T$,

$$\|\tilde{\mathbf{P}} - \mathbf{P}\tilde{\mathbf{P}}\| \leq \frac{\mu}{\Delta - \mu}. \tag{7}$$

We next show that the spectrum of $\tilde{\mathbf{A}}$ also has a gap at $k$. Using Weyl's inequality,

$$\tilde{\sigma}_k^2 \geq \sigma_k^2 - \mu\sigma_1^2 \text{ and } \tilde{\sigma}_{k+1}^2 \leq \sigma_{k+1}^2 + \mu\sigma_1^2$$

Hence using Equation (6)

$$\tilde{\sigma}_k^2 - \tilde{\sigma}_{k+1}^2 \geq \sigma_{k+1}^2 - \sigma_k^2 - 2\mu\sigma_1^2 \geq (\Delta - 2\mu)\sigma_1^2.$$

So we apply Wedin's theorem to $\tilde{\mathbf{A}}$ to get

$$\|\mathbf{P} - \tilde{\mathbf{P}}\mathbf{P}\| \leq \frac{\mu}{\Delta - 3\mu}. \tag{8}$$

Plugging (7) and (8) into (5),

$$\|\mathbf{P} - \tilde{\mathbf{P}}\|^2 \leq \frac{\mu}{\Delta - \mu} + \frac{\mu}{\Delta - 3\mu} \leq \frac{2\mu}{\Delta - 3\mu} \leq \frac{4\mu}{\Delta}$$

where the last inequality is becuase $\Delta - 3\mu \geq \Delta/2$ since we assumed $\Delta \geq 6\mu$. The claim follows by taking square roots on both sides. ∎

We now move to proving bounds for items (2) and (3), which are given by Theorems 4 and 6 respectively.

**Theorem 4.** *Let* $\mu \leq \min(\Delta^3 k^2, 1/(20k))$.

$$\|\mathbf{U}_k\mathbf{\Sigma}_k^2\mathbf{U}_k^T - \tilde{\mathbf{U}}_k\mathbf{\Sigma}_k^2\tilde{\mathbf{U}}_k^T\| \leq 8\sigma_1^2(\mu k)^{1/3}.$$

The full proof of the theorem is quite technical and is stated in Section E. Here we prove a special case that captures the main idea. The simplifying assumption we make is that the values of the diagonal matrix in the operator are distinct and well separated. In the full proof we decompose $\mathbf{\Sigma}_k$ to a well separated component and a small residual component.

**Definition 5.** $\mathbf{\Lambda} = \mathrm{diag}(\lambda_1, \ldots \lambda_k)$ *is* *monotone and* $\delta$-*well separated if*

- $\lambda_{i+1} \geq \lambda_i$ for $1 \leq i < k$.

- The $\lambda$s could be partitioned to buckets so that all values in the same buckets are equal, and values across buckets are well separated. Formally, $1 \ldots k$ are partitioned to $m$ buckets $B_1, \ldots B_m$ so that if $i, j \in B_\ell$ then $\lambda_i = \lambda_j$. Yet, if $i \in B_\ell$ and $j \in B_{\ell+1}$ then $\lambda_i - \lambda_j > \delta\lambda_1$.

The idea is to show $\mathbf{\Sigma} = \mathbf{\Lambda} + \mathbf{\Omega}$ where $\mathbf{\Lambda}$ is monotone and well separated and $\mathbf{\Omega}$ has small norm. The next two lemmas handle these two components. We first state a simple lemma which handles the case where $\mathbf{\Omega}$ has small norm.

**Lemma 4.** *For any diagonal matrix* $\mathbf{\Omega}$,

$$\|\mathbf{U}_k\mathbf{\Omega}\mathbf{U}_k^T - \tilde{\mathbf{U}}_k\mathbf{\Omega}\tilde{\mathbf{U}}_k^T\| \leq 2\|\mathbf{\Omega}\|.$$

**Proof.** By the triangle inequality, $\|\mathbf{U}_k\mathbf{\Omega}\mathbf{U}_k^T - \tilde{\mathbf{U}}_k\mathbf{\Omega}\tilde{\mathbf{U}}_k^T\| \leq \|\mathbf{U}_k\mathbf{\Omega}\mathbf{U}_k^T\| + \|\tilde{\mathbf{U}}_k\mathbf{\Omega}\tilde{\mathbf{U}}_k^T\|$. The bound follows as $\mathbf{U}_k$ and $\tilde{\mathbf{U}}$ are orthonormal matrices. ∎

We next state the result for the case where $\mathbf{\Lambda}$ is monotone and well separated. In order to prove this result, we need the following direct corollary of Lemma 3:

**Lemma 5.** *For all* $j \leq m$,

$$\|\mathbf{U}_{b_j}\mathbf{U}_{b_j}^T - \tilde{\mathbf{U}}_{b_j}\tilde{\mathbf{U}}_{b_j}^T\| \leq \sqrt{\frac{2\mu\sigma_1^2}{\delta\sigma_1^2 - 3\mu\sigma_1^2}} \leq 2\sqrt{\frac{\mu}{\delta}}. \tag{9}$$

Using this, we now proceed as follows.

**Lemma 6.** *Let* $6\mu \leq \delta \leq \Delta$. *Let* $\mathbf{\Lambda} = \mathrm{diag}(\lambda_1, \dots \lambda_k)$ *be a monotone and* $\delta\sigma_1^2$ *well separated diagonal matrix. Then*

$$\|\mathbf{U}_k\mathbf{\Lambda}\mathbf{U}_k^T - \tilde{\mathbf{U}}_k\mathbf{\Lambda}\tilde{\mathbf{U}}_k^T\| \leq 2\|\mathbf{\Lambda}\|\sqrt{\frac{\mu}{\delta}}.$$

**Proof.** We denote by $b_j$ the largest index that falls in bucket $B_j$. Let us set $\lambda_{B_{m+1}} = 0$ for convenience. Since

$$\mathbf{U}_{b_j}\mathbf{U}_{b_j}^T = \sum_{i=1}^{b_j} u^{(i)}u^{(i)^T},$$

we can write,

$$\mathbf{U}_k\mathbf{\Lambda}\mathbf{U}_k^T = \sum_{j=1}^m (\lambda_{b_j} - \lambda_{b_{j+1}})\mathbf{U}_{b_j}\mathbf{U}_{b_j}^T$$

and similarly for $\tilde{\mathbf{U}}_k\mathbf{\Lambda}\tilde{\mathbf{U}}_k^T$. So we can write

$$\mathbf{U}_k\mathbf{\Lambda}\mathbf{U}_k^T - \tilde{\mathbf{U}}_k\mathbf{\Lambda}\tilde{\mathbf{U}}_k^T = \sum_{j=1}^m (\lambda_{b_j} - \lambda_{b_{j+1}})(\mathbf{U}_{b_j}\mathbf{U}_{b_j}^T - \tilde{\mathbf{U}}_{b_j}\tilde{\mathbf{U}}_{b_j}^T).$$

Therefore by using Lemma 5,

$$\|\mathbf{U}_k\mathbf{\Lambda}\mathbf{U}_k^T - \tilde{\mathbf{U}}_k\mathbf{\Lambda}\tilde{\mathbf{U}}_k^T\| = \sum_{j=1}^m |\lambda_{b_j} - \lambda_{b_{j+1}}|\|(\mathbf{U}_{b_j}\mathbf{U}_{b_j}^T - \tilde{\mathbf{U}}_{b_j}\tilde{\mathbf{U}}_{b_j}^T)\| \leq 2\sqrt{\frac{\mu}{\delta}}\sum_{j=1}^m |\lambda_{b_j} - \lambda_{b_{j+1}}|,$$

where the second inequality is by the triangle inequality and by applying Lemma 4. Thus proving that $\sum_{j=1}^m |\lambda_{b_j} - \lambda_{b_{j+1}}| \leq \|\mathbf{\Lambda}\|$ would imply the claim. Indeed

$$\sum_{j=1}^k |\lambda_{b_j} - \lambda_{b_{j+1}}| = \sum_{j=1}^k (\lambda_{b_j} - \lambda_{b_{j+1}}) = \lambda_{b_1} - \lambda_{b_{k+1}} \leq \|\mathbf{\Lambda}\|.$$

∎

Note that though Lemma 6 assumes that the eigenvalues in each bucket are equal, the key step where we apply Wedin's theorem (Lemma 3) only uses the fact that there is some separation between the eigenvalues in different buckets. Lemma 10 in the appendix does this generalization by relaxing the assumption that the eigenvalues in each bucket are equal. The final proof of Theorem 4 works by splitting the eigenvalues into different buckets or intervals such that all the eigenvalues in the same interval have small separation, and the eigenvalues in different intervals have large separation. We then use Lemma 10 and Lemma 4 along with the triangle inequality to bound the perturbation due to the well-separated part and the residual part respectively.

The bound corresponding to (3) is given in following theorem:

**Theorem 6.** *Let $\kappa_k$ denote the condition number $\kappa_k = \sigma_1^2/\sigma_k^2$. Let $\mu \leq \min(\Delta^3(k\kappa_k)^2, 1/(20k\kappa_k))$. Then,*

$$\|\mathbf{U}_k\boldsymbol{\Sigma}_k^{-2}\mathbf{U}_k^T - \tilde{\mathbf{U}}_k\boldsymbol{\Sigma}_k^{-2}\tilde{\mathbf{U}}_k^T\| \leq \frac{8}{\sigma_k^2}(\mu k\kappa_k)^{1/3}.$$

The proof uses similar ideas to those of Theorem 4 and is deferred to Section E.

## B  Pointwise guarantees for Anomaly Detection

Let the input $\mathbf{A} \in \mathbb{R}^{n \times d}$ have SVD $\mathbf{U}\boldsymbol{\Sigma}\mathbf{V}^T$ be its SVD. Write $a_{(i)}$ in the basis of right singular vectors as $a_{(i)} = \sum_{j=1}^{d} \alpha_j v^{(j)}$. Recall that we defined its rank-$k$ leverage score and projection distance respectively as

$$L^k(i) = \sum_{j=1}^{k} \frac{\alpha_j^2}{\sigma_j^2} = \|\boldsymbol{\Sigma}_k^{-1}\mathbf{V}_k^T a_{(i)}\|_2^2, \tag{10}$$

$$T^k(i) = \sum_{j=k+1}^{d} \alpha_j^2 = a_{(i)}^T(\mathbf{I} - \mathbf{V}_k\mathbf{V}_k^T)a_{(i)}. \tag{11}$$

To approximate these scores, it is natural to use a row-space approximation, or rather a sketch $\tilde{\mathbf{A}} \in R^{\ell \times d}$ that approximates the covariance matrix $\mathbf{A}^T\mathbf{A}$ as below:

$$\|\mathbf{A}^T\mathbf{A} - \tilde{\mathbf{A}}^T\tilde{\mathbf{A}}\| \leq \mu\sigma_1^2. \tag{12}$$

Given such a sketch, our approximation is the following: compute $\tilde{\mathbf{A}} = \tilde{\mathbf{U}}\tilde{\boldsymbol{\Sigma}}\tilde{\mathbf{V}}^t$. The estimates for $L^k(i)$ and $T^k(i)$ respectively are

$$\tilde{L}^k(i) = \|\tilde{\boldsymbol{\Sigma}}_k^{-1}\tilde{\mathbf{V}}_k^T a_{(i)}\|_2^2, \tag{13}$$

$$\tilde{T}^k(i) = a_{(i)}^T(\mathbf{I} - \tilde{\mathbf{V}}_k\tilde{\mathbf{V}}_k^T)a_{(i)}. \tag{14}$$

Given the sketch $\tilde{\mathbf{A}}$, we expect that $\tilde{\mathbf{V}}_k$ is a good approximation to the row space spanned by $\mathbf{V}_k$, since the covariance matrices of the rows are close. In contrast the columns spaces of $\mathbf{A}$ and $\tilde{\mathbf{A}}$ are hard to compare since they lie in $\mathbb{R}^n$ and $\mathbb{R}^\ell$ respectively. The closeness of the row spaces follows from the results from Section A but applied to $\mathbf{A}^T$ rather than $\mathbf{A}$ itself. The results there require that $\|\mathbf{A}\mathbf{A}^T - \tilde{\mathbf{A}}\tilde{\mathbf{A}}^T\|$ is small, and Equation (12) implies that this assumption holds for $\mathbf{A}^T$.

We first state our approximation guarantees for $T^k$.

**Theorem 7.** *Assume that $\mathbf{A}$ is $(k, \Delta)$-separated. Let $\varepsilon < 1/3$ and let $\tilde{\mathbf{A}}$ satisfy Equation (12) for $\mu = \varepsilon^2\Delta$. Then for every $i$,*

$$|T^k(i) - \tilde{T}^k(i)| \leq \varepsilon\|a_{(i)}\|_2^2.$$

**Proof.**  We have

$$\begin{aligned}
|T^k(i) - \tilde{T}^k(i)| &= |a_{(i)}^T(\mathbf{I} - \mathbf{V}_k\mathbf{V}_k^T)a_{(i)} - a_{(i)}^T(\mathbf{I} - \tilde{\mathbf{V}}_k\tilde{\mathbf{V}}_k^T)a_{(i)}| \\
&= |a_{(i)}(\tilde{\mathbf{V}}_k\tilde{\mathbf{V}}_k^T - \mathbf{V}_k\mathbf{V}_k^T)a_{(i)}| \\
&\leq \|\tilde{\mathbf{V}}_k\tilde{\mathbf{V}}_k^T - \mathbf{V}_k\mathbf{V}_k^T\|\|a_{(i)}\|_2^2 \\
&\leq \varepsilon\|a_{(i)}\|_2^2
\end{aligned}$$

where in the last line we use Lemma 3, applied to the projection onto columns of $\mathbf{A}^T$, which are the rows of $\mathbf{A}$. The condition $\mu < \Delta/6$ holds since $\mu = \varepsilon^2\Delta$ for $\varepsilon < 1/3$.  ∎

How meaningful is the above additive approximation guarantee? For each row, the additive error is $\varepsilon\|a_{(i)}\|_2^2$. It might be that $T^k(i) \ll \varepsilon\|a_{(i)}\|_2^2$ which happens when the row is almost entirely within the principal subspace. But in this case, the points are not anomalies, and we have $\tilde{T}^k(i) \leq 2\varepsilon\|a_{(i)}\|_2^2$, so these points will not seem anomalous from the approximation either. The interesting case is when

$T^k(i) \geq \beta \|a_{(i)}\|_2^2$ for some constant $\beta$ (say $1/2$). For such points, we have $\tilde{T}^k(i) \in (\beta \pm \epsilon)\|a_{(i)}\|_2^2$, so we indeed get multiplicative approximations.

Next we give our approximation guarantee for $L^k$, which relies on the perturbation bound in Theorem 6.

**Theorem 8.** *Assume that $\mathbf{A}$ is $(k, \Delta)$-separated. Let $\tilde{\mathbf{A}}$ be as in Equation* (12). *Let*

$$\varepsilon \leq \min\left(\kappa_k \Delta, \frac{1}{k}\right)\mathrm{sr}(\mathbf{A})\kappa_k, \quad \mu = \frac{\varepsilon^3 k^2}{10^3 \mathrm{sr}(A)^3 \kappa_k^4}.$$

*Then for every $i$,*

$$|L^k(i) - \tilde{L}^k(i)| \leq \varepsilon k \frac{\|a_{(i)}\|_2^2}{\|\mathbf{A}\|_F^2}. \tag{15}$$

**Proof.** Since $L^k(i) = \|\mathbf{\Sigma}_k^{-1}\mathbf{V}_k^T a_{(i)}\|_2^2$, $\tilde{L}^k(i) = \|\tilde{\mathbf{\Sigma}}_k^{-1}\tilde{\mathbf{V}}_k^T a_{(i)}\|_2^2$, it will suffice to show that

$$\|\mathbf{V}_k \mathbf{\Sigma}_k^{-2}\mathbf{V}_k^T - \tilde{\mathbf{V}}_k \tilde{\mathbf{\Sigma}}^{-2}\tilde{\mathbf{V}}_k^T\| \leq \frac{\varepsilon k}{\|\mathbf{A}\|_F^2}. \tag{16}$$

To prove inequality (16), we will bound the LHS as

$$\|\mathbf{V}_k \mathbf{\Sigma}^{-2}\mathbf{V}_k^T - \tilde{\mathbf{V}}_k \tilde{\mathbf{\Sigma}}^{-2}\tilde{\mathbf{V}}_k^T\| \leq \|\mathbf{V}_k \mathbf{\Sigma}_k^{-2}\mathbf{V}_k^T - \tilde{\mathbf{V}}_k^T \mathbf{\Sigma}_k^{-2}\tilde{\mathbf{V}}_k^T\| + \|\tilde{\mathbf{V}}_k(\mathbf{\Sigma}^{-2} - \tilde{\mathbf{\Sigma}}^{-2})\tilde{\mathbf{V}}_k^T\| \tag{17}$$

For the first term, we apply Theorem 6 to $\mathbf{A}^T$ to get

$$\|\mathbf{V}_k \mathbf{\Sigma}^{-2}\mathbf{V}_k^T - \tilde{\mathbf{V}}_k^T \mathbf{\Sigma}_k^{-2}\tilde{\mathbf{V}}_k^T\| \leq \frac{8}{\sigma_k^2}(\mu k \kappa_k)^{1/3}. \tag{18}$$

We bound the second term as

$$\|\tilde{\mathbf{V}}_k(\mathbf{\Sigma}^{-2} - \tilde{\mathbf{\Sigma}}^{-2})\tilde{\mathbf{V}}_k^T\| = \max_{i \in [k]} |\sigma_i^{-2} - \tilde{\sigma}_i^{-2}| = \max_{i \in [k]} \frac{|\sigma_i^2 - \tilde{\sigma}_i^2|}{\sigma_i^2 \tilde{\sigma}_i^2} \leq \frac{\mu \sigma_1^2}{\sigma_k^2 \tilde{\sigma}_k^2} \tag{19}$$

where we use Weyl's inequality to bound $\sigma_i^2 - \tilde{\sigma}_i^2$. Using Weyl's inequality and the fact that $\mu \geq 1/(20k\kappa_k)$,

$$\tilde{\sigma}_k^2 \geq \sigma_k^2 - \mu\sigma_1^2 \geq \sigma_k^2 - \sigma_k^2/10k \geq \sigma_k^2/2,$$
$$\frac{\mu\sigma_1^2}{\sigma_k^2 \tilde{\sigma}_k^2} \leq \frac{2\mu\sigma_1^2}{\sigma_k^4} = \frac{2\mu\kappa_k}{\sigma_k^2} \tag{20}$$

Plugging Equations (18) and (20) into Equation (17) gives

$$\|\mathbf{V}_k \mathbf{\Sigma}^{-2}\mathbf{V}_k^T - \tilde{\mathbf{V}}_k \tilde{\mathbf{\Sigma}}^{-2}\tilde{\mathbf{V}}_k^T\| \leq \frac{1}{\sigma_k^2}(8(\mu k\kappa_k)^{1/3} + 2\mu\kappa_k) \leq \frac{10}{\sigma_k^2}(\mu k\kappa_k)^{1/3} \tag{21}$$

$$\leq \frac{10}{\sigma_k^2}\left(\frac{\varepsilon^3 k^3}{10^3 \mathrm{sr}(\mathbf{A})^3 \kappa_k^3}\right)^{1/3} \leq \frac{k\varepsilon}{\sigma_k^2 \kappa_k \mathrm{sr}(\mathbf{A})} = \frac{k\varepsilon}{\|\mathbf{A}\|_F^2}.$$

Equation (21) follows by Theorem 6. The conditions on $\mu$ needed to apply it are guaranteed by our choice of $\mu$ and $\varepsilon$. ∎

To interpret this guarantee, consider the setting when all the points have roughly the same 2-norm. More precisely, if for some constant $C$

$$\frac{\|a_{(i)}\|_2^2}{\|\mathbf{A}\|_F^2} \leq \frac{C}{i}$$

then Equation (15) gives

$$|L^k(i) - \tilde{L}^k(i)| \leq C\varepsilon k/n.$$

Note that $k$ is a constant whereas $n$ grows as more points come in. As mentioned in the discussion following Theorem 1, the points which are considered outliers are those where $L^k(i) \gg \frac{k}{n}$. For the parameters setting, if we let $\kappa_k = O(1)$ and $\mathrm{sr}(A) = \Theta(k)$, then our bound on $\epsilon$ reduces to $\varepsilon \leq \min(\Delta, 1/k)$, and our choice of $\mu$ reduces to $\mu \approx \varepsilon^3/k$.

To efficiently compute a sketch that satisfies (12), we can use Frequent Directions [18]. We use the improved analysis of Frequent Directions in Ghashami et al. [26]:

**Theorem 9.** *[26] There is an algorithm that takes the rows of $\mathbf{A}$ in streaming fashion and computes a sketch $\tilde{\mathbf{A}} \in \mathbb{R}^{\ell \times d}$ satisfying Equation (12) where $\ell = \sum_{i=k+1}^{n} \sigma_i^2 / (\sigma_1^2 \mu)$.*

Let $\tilde{\mathbf{A}} = \tilde{\mathbf{U}} \tilde{\mathbf{\Sigma}} \tilde{\mathbf{V}}^T$. The algorithm maintains $\tilde{\mathbf{\Sigma}}$, $\tilde{\mathbf{V}}$. It uses $O(d\ell)$ memory and requires time at most $O(d\ell^2)$ per row. The total time for processing $n$ rows is $O(nd\ell)$. If $\ell \ll d$, this is a significant improvement over the naive algorithm in both memory and time. If we use Frequent directions, we set

$$\ell = \frac{\sum_{i=k+1}^{d} \sigma_i^2}{\sigma_1^2 \mu} \tag{22}$$

where $\mu$ is set according to Theorem 7 and 8. This leads to $\ell = \text{poly}(k, \text{sr}(\mathbf{A}), \kappa_k, \Delta, \epsilon^{-1})$. Note that this does not depend on $d$, and hence is considerably smaller for our parameter settings of interest.

### B.1 The Online Setting

We now consider the online scenario where the leverage scores and projection distances are defined only with respect to the input seen so far. Consider again the motivating example where each machine in a data center produces streams of measurements. Here, it is desirable to determine the anomaly score of a data point online as it arrives, with respect to the data produced so far, and by a streaming algorithm. We first define the online anomaly measures. Let $\mathbf{A} \in \mathbb{R}^{(i-1) \times d}$ denote the matrix of points that have arrived so far (excluding $a_{(i)}$) and let $\mathbf{U} \mathbf{\Sigma} \mathbf{V}^T$ be its SVD. Write $a_{(i)}$ in the basis of right singular vectors as $a_{(i)} = \sum_{j=1}^{d} \alpha_j v^{(j)}$. We define its rank-$k$ leverage score and projection distance respectively as

$$l^k(i) = \sum_{j=1}^{k} \frac{\alpha_j^2}{\sigma_j^2} = \|\mathbf{\Sigma}_k^{-1} \mathbf{V}_k^T a_{(i)}\|_2^2, \tag{23}$$

$$t^k(i) = \sum_{j=k+1}^{d} \alpha_j^2 = a_{(i)}^T (\mathbf{I} - \mathbf{V}_k \mathbf{V}_k^T) a_{(i)}. \tag{24}$$

Note that in the online setting there is a one-pass streaming algorithm that can compute both these scores, using time $O(d^3)$ per row and $O(d^2)$ memory. This algorithm maintains the $d \times d$ covariance matrix $\mathbf{A}^T \mathbf{A}$ and computes its SVD to get $\mathbf{V}_k, \mathbf{\Sigma}_k$. From these, it is easy to compute both $l^k$ and $t^k$.

All our guarantees from the previous subsection directly carry over to this online scenario, allowing us to significantly improve over this baseline. This is because the guarantees are pointwise, hence they also hold for every data point if the scores are only defined for the input seen so far. This implies a one-pass algorithm which can approximately compute the anomaly scores (i.e., satisfies the guarantees in Theorem 7 and 8) and uses space $O(d\ell)$ and requires time $O(nd\ell)$ for $\ell = \text{poly}(k, \text{sr}(\mathbf{A}), \kappa_k, \Delta, \epsilon^{-1})$ (independent of $d$).

The $\Omega(d)$ lower bounds in Section D show that one cannot hope for sublinear dependence on $d$ for pointwise estimates. In the next section, we show how to eliminate the dependence on $d$ in the space requirement of the algorithm in exchange for weaker guarantees.

## C  Average-case guarantees for Anomaly Detection

In this section, we present an approach which circumvents the $\Omega(d)$ lower bounds by relaxing the pointwise approximation guarantee.

Let $\mathbf{A} = \mathbf{U} \mathbf{\Sigma} \mathbf{V}^T$ be the SVD of $\mathbf{A}$. The outlier scores we wish to compute are

$$L^k(i) = \|e_i^T \mathbf{U}_k\|_2^2 = \|\mathbf{\Sigma}_k^{-1} \mathbf{V}_k^T a_{(i)}\|_2^2, \tag{25}$$

$$T^k(i) = \|a_{(i)}\|_2^2 - \|e_i^T \mathbf{U}_k \mathbf{\Sigma}_k\|_2^2 = \|a_{(i)}\|_2^2 - \|\mathbf{V}_k^T a_{(i)}\|_2^2. \tag{26}$$

Note that these scores are defined with respect to the principal space of the entire matrix. We present a guarantee for any sketch $\tilde{\mathbf{A}} \in R^{n \times \ell}$ that approximates the column space of $\mathbf{A}$, or equivalently the covariance matrix $\mathbf{A} \mathbf{A}^T$ of the row vectors. We can work with any sketch $\mathbf{A}$ where

$$\|\mathbf{A} \mathbf{A}^T - \tilde{\mathbf{A}} \tilde{\mathbf{A}}^T\| \leq \mu \sigma_1^2. \tag{27}$$

Theorem 10 stated in Section C.2 shows that such a sketch can be obtained for instance by a random projection $\mathbf{R}$ onto $\mathbb{R}^\ell$ for $\ell = \mathrm{sr}(\mathbf{A})/\mu^2$: let $\tilde{\mathbf{A}} = \mathbf{A}\mathbf{R}$ for $\mathbf{R} \in \mathbb{R}^{d \times \ell}$ chosen from an appropriately family of random matrices. However, we need to be careful in our choice of the family of random matrices, as naively storing a $(d \times \ell)$ matrix requires $O(d\ell)$ space, which would increase the space requirement of our streaming algorithm. For example, if we were to choose each entry of $\mathbf{R}$ to be i.i.d. be $\pm\frac{1}{\sqrt{\ell}}$, then we would need to store $O(d\ell)$ random bits corresponding to each entry of $\mathbf{R}$.

But Theorem 10 also shows that this is unnecessary and we do not need to explicitly store a $(d \times \ell)$ random matrix. The guarantees of Theorem 10 also hold when $\mathbf{R}$ is a pseudorandom matrix with the entries being $\pm\frac{1}{\sqrt{\ell}}$ with $\log(\mathrm{sr}(\mathbf{A})/\delta)$-wise independence instead of full independence . Therefore, by using a simple polynomial based hashing scheme [28] we can get the same sketching guarantees using only $O(\log(d)\log(\mathrm{sr}(\mathbf{A})/\delta))$ random bits and hence only $O(\log(d)\log(\mathrm{sr}(\mathbf{A})/\delta))$ space. Note that each entry of $\mathbf{R}$ can be computed from this random seed in time $O(\log(d)\log(\mathrm{sr}(\mathbf{A})/\delta))$.

Theorem 11 stated in Section C.3 shows that such a sketch can also be obtained for a length-squared sub-sampling of the columns of the matrix, for $\ell = \tilde{O}(\mathrm{sr}(\mathbf{A})/\mu^2)$ (where the $\tilde{O}$ hides logarithmic factors).

Given such a sketch, we expect $\tilde{\mathbf{U}}_k$ to be a good approximation to $\mathbf{U}_k$. So we define our approximations in the natural way:

$$\tilde{L}^k(i) = \|e_i^T \tilde{\mathbf{U}}_k^i\|_2^2, \tag{28}$$

$$\tilde{T}^k(i) = \|a_{(i)}\|_2^2 - \|e_i^T \tilde{\mathbf{U}}_k \tilde{\mathbf{\Sigma}}_k\|_2^2. \tag{29}$$

The analysis then relies on the machinery from Section A. However, $\tilde{\mathbf{U}}_k$ lies in $\mathbb{R}^{n \times k}$ which is too costly to compute and store, whereas $\tilde{\mathbf{V}}_k$ in contrast lies in $\mathbb{R}^{\ell \times k}$, for $\ell = 1/\mu^2 \max(\mathrm{sr}(\mathbf{A}), \log(1/\delta))$. In particular, both $\ell$ and $k$ are independent of $n, d$ and could be significantly smaller. In many settings of practical interest, we have $\mathrm{sr}(\mathbf{A}) \approx k$ and both are constants independent of $n, d$. So in our algorithm, we use the following equivalent definition in terms of $\tilde{\mathbf{V}}_k, \tilde{\mathbf{\Sigma}}_k$.

$$\tilde{L}^k(i) = \|\tilde{\mathbf{\Sigma}}_k^{-1}\tilde{\mathbf{V}}_k^T(\mathbf{R}^T a_{(i)})\|_2^2, \tag{30}$$

$$\tilde{T}^k(i) = \|a_{(i)}\|_2^2 - \|\tilde{\mathbf{V}}_k^T(\mathbf{R}^T a_{(i)})\|_2^2. \tag{31}$$

For the random projection algorithm, we compute $\tilde{\mathbf{A}}^T \tilde{\mathbf{A}}$ in $\mathbb{R}^{\ell \times \ell}$ in the first pass, and then run SVD on it to compute $\tilde{\mathbf{\Sigma}}_k \in \mathbb{R}^{k \times k}$ and $\tilde{\mathbf{V}}_k \in \mathbb{R}^{\ell \times k}$. Then in the second pass, we use these to we compute $\tilde{L}^k$ and $\tilde{T}^k$. The total memory needed in the first pass is $O(\ell^2)$ for the covariance matrix. In the second pass, we need $O(k\ell)$ memory for storing $\mathbf{V}_k$. We also need $O(\log(d)\log(\mathrm{sr}(\mathbf{A})/\delta))$ additional memory for storing the random seed from which the entries of $\mathbf{R}$ can be computed efficiently.

## C.1 Our approximation guarantees

We now turn to the guarantees. Given the $\Omega(d)$ lower bound from Section D, we cannot hope for a strong pointwise guarantee, rather we will show a guarantee that hold on average, or for most points.

The following simple Lemma bounds the sum of absolute values of diagonal entries in symmetric matrices.

**Lemma 7.** *Let $\mathbf{A} \in \mathbb{R}^{n \times n}$ be symmetric. Then*

$$\sum_{i=1}^n \left|\mathbf{e}_i^T \mathbf{A} \mathbf{e}_i\right| \leq \mathrm{rank}(\mathbf{A})\|\mathbf{A}\|.$$

**Proof.** Consider the eigenvalue decomposition of $\mathbf{A} = \mathbf{Q}\mathbf{\Lambda}\mathbf{Q}^T$ where $\mathbf{\Lambda}$ is the diagonal matrix of eigenvalues of $\mathbf{A}$, and $\mathbf{Q}$ has orthonormal columns, so $\|\mathbf{Q}\|_F^2 = \mathrm{rank}(\mathbf{A})$. We can write,

$$\sum_{i=1}^n \left|\mathbf{e}_i^T \mathbf{A} \mathbf{e}_i\right| = \sum_{i=1}^n \left|\mathbf{e}_i^T \mathbf{Q}\mathbf{\Lambda}\mathbf{Q}^T \mathbf{e}_i\right| = \sum_{i=1}^n \sum_{j=1}^n \left|\mathbf{\Lambda}_{i,i}\mathbf{Q}_{i,j}^2\right|$$

$$\leq \|\mathbf{A}\| \sum_{i,j=1}^n \left|\mathbf{Q}_{i,j}^2\right| = \|\mathbf{A}\|\|\mathbf{Q}\|_F^2 = \mathrm{rank}(\mathbf{A})\|\mathbf{A}\|.$$

■

We first state and prove Lemma 8 which bounds the average error in estimating $L^k$.

**Lemma 8.** *Assume that* $\mathbf{A}$ *is* $(k, \Delta)$*-separated. Let* $\tilde{\mathbf{A}}$ *satisfy Equation* (27) *for* $\mu = \varepsilon^2 \Delta / 16$ *where* $\varepsilon < 1$. *Then*

$$\sum_{i=1}^{n} |L^k(i) - \tilde{L}^k(i)| \leq \varepsilon \sum_{i=1}^{n} L^k(i).$$

**Proof.** By Equations (25) and (28)

$$\sum_{i=1}^{n} |L^k(i) - \tilde{L}^k(i)| = \sum_{i=1}^{n} |e_i^t \mathbf{U}_k \mathbf{U}_k^T e_i - e_i^t \tilde{\mathbf{U}}_k \tilde{\mathbf{U}}_k^T e_i|.$$

Let $\mathbf{C} = \mathbf{U}_k \mathbf{U}_k^T - \tilde{\mathbf{U}}_k \tilde{\mathbf{U}}_k^T$, so that $\text{rank}(\mathbf{C}) \leq 2k$. By Lemma 3 (which applies since $\mu \leq \Delta/16$), we have

$$\|\mathbf{C}\| \leq 2\sqrt{\frac{\mu}{\Delta}} \leq \frac{\varepsilon}{2}.$$

So applying Lemma 7, we get

$$\sum_{i=1}^{n} |e_i^t \mathbf{U}_k \mathbf{U}_k^T e_i - e_i^t \tilde{\mathbf{U}}_k \tilde{\mathbf{U}}_k^T e_i| \leq \frac{\varepsilon}{2} 2k = \varepsilon k.$$

The claim follows by noting that the columns of $\mathbf{U}_k$ are orthonormal, so $\sum_{i=1}^{n} L^k(i) = k$. ■

The guarantee above shows that the average additive error in estimating $L^k(i)$ is $\frac{\varepsilon}{n} \sum_{i=1}^{n} L^k(i)$ for a suitable $\varepsilon$. Note that the average value of $L^k(i)$ is $\frac{1}{n} \sum_{i=1}^{n} L^k(i)$, hence we obtain small additive errors on average. Additive error guarantees for $L^k(i)$ translate to multiplicative guarantees as long as $L^k(i)$ is not too small, but for outlier detection the candidate outliers are those points for which $L^k(i)$ is large, hence additive error guarantees are meaningful for preserving outlier scores for points which could be outliers.

Similarly, Lemma 9 bounds the average error in estimating $T^k$.

**Lemma 9.** *Assume that* $\mathbf{A}$ *is* $(k, \Delta)$*-separated. Let*

$$\varepsilon \leq \frac{\min(\Delta k^2, k)}{\text{sr}(A)}. \tag{32}$$

*Let* $\tilde{\mathbf{A}}$ *satisfy Equation* (27) *for*

$$\mu = \frac{\varepsilon^3 \text{sr}(\mathbf{A})^3}{125 k^4}. \tag{33}$$

*Then*

$$\sum_{i=1}^{n} |T^k(i) - \tilde{T}^k(i)| \leq \frac{\varepsilon \|\mathbf{A}\|_F^2}{\|\mathbf{A} - \mathbf{A}_k\|_F^2} \sum_{i=1}^{n} T^k(i).$$

**Proof.** By Equations (26) and (29), we have

$$\sum_{i=1}^{n} |T^k(i) - \tilde{T}^k(i)| = \sum_{i=1}^{n} |\|e_i^T \mathbf{U}_k \boldsymbol{\Sigma}_k\|_2^2 - \|e_i^T \tilde{\mathbf{U}}_k \tilde{\boldsymbol{\Sigma}}_k\|_2^2| = \sum_{i=1}^{n} |e_i^t \mathbf{U}_k \boldsymbol{\Sigma}_k^2 \mathbf{U}_k^T e_i - e_i^t \tilde{\mathbf{U}}_k \tilde{\boldsymbol{\Sigma}}_k^2 \tilde{\mathbf{U}}_k^T e_i|$$

Let $\mathbf{C} = \mathbf{U}_k \boldsymbol{\Sigma}_k^2 \mathbf{U}_k^T - \tilde{\mathbf{U}}_k \tilde{\boldsymbol{\Sigma}}_k^2 \tilde{\mathbf{U}}_k^T$. Then $\text{rank}(\mathbf{C}) \leq 2k$. We now bound its operator norm as follows

$$\|\mathbf{C}\| \leq \|\mathbf{U}_k \boldsymbol{\Sigma}_k^2 \mathbf{U}_k^T - \tilde{\mathbf{U}}_k \boldsymbol{\Sigma}_k^2 \tilde{\mathbf{U}}_k^T\| + \|\tilde{\mathbf{U}}_k (\boldsymbol{\Sigma}_k^2 - \tilde{\boldsymbol{\Sigma}}_k^2) \tilde{\mathbf{U}}_k^T\|.$$

To bound the first term, we use Theorem 4 (the condition on $\mu$ holds by our choice of $\varepsilon$ in Equation (32) and $\mu$ in Equation (33)) to get

$$\|\mathbf{U}_{\leq k} \boldsymbol{\Sigma}_k^2 \mathbf{U}_{\leq k}^t - \tilde{\mathbf{U}}_{\leq k} \boldsymbol{\Sigma}_k^2 \tilde{\mathbf{U}}_{\leq k}^t\| \leq 4\sigma_1(\mathbf{A})^2 (\mu k)^{1/3}.$$

For the second term, we use

$$\|\tilde{\mathbf{U}}_k(\boldsymbol{\Sigma}_k^2 - \tilde{\boldsymbol{\Sigma}}_k^2)\tilde{\mathbf{U}}_k^T\| \leq \|\boldsymbol{\Sigma}_k^2 - \tilde{\boldsymbol{\Sigma}}_k^2\| \leq \mu\sigma_1(\mathbf{A})^2.$$

Overall, we get $\|\mathbf{C}\| \leq 5\sigma(\mathbf{A})^2(\mu k)^{1/3}$. So applying Lemma 7, we get

$$\sum_{i=1}^{n} |e_i^t \mathbf{U}_k \boldsymbol{\Sigma}_k^2 \mathbf{U}_k^T e_i - e_i^t \tilde{\mathbf{U}}_k \tilde{\boldsymbol{\Sigma}}_k^2 \tilde{\mathbf{U}}_k^T e_i| \leq 5\sigma(\mathbf{A})^2(\mu k)^{1/3} \cdot 2k$$

$$\leq 5\sigma(\mathbf{A})^2 \mu^{1/3} k^{4/3}$$

$$\leq \varepsilon \mathrm{sr}(\mathbf{A})\sigma(\mathbf{A})^2 = \varepsilon\|\mathbf{A}\|_{\mathrm{F}}^2.$$

In order to obtain the result in the form stated in the Lemma, note that $\sum_{i=1}^{n} T^k(i) = \|\mathbf{A} - \mathbf{A}_k\|_{\mathrm{F}}^2$. ∎

Typically, we expect $\|\mathbf{A} - \mathbf{A}_k\|_{\mathrm{F}}^2$ to be a constant fraction of $\|\mathbf{A}\|_{\mathrm{F}}^2$. Hence the guarantee above says that on average, we get good additive guarantees.

## C.2 Guarantees for random projections

**Theorem 10.** *[29, 30] Consider any matrix $\mathbf{A} \in \mathbb{R}^{n \times d}$. Let $\mathbf{R} = (1/\sqrt{\ell})\mathbf{X}$ where $\mathbf{X} \in \mathbb{R}^{d \times \ell}$ is a random matrix drawn from any of the following distributions of matrices. Let $\tilde{\mathbf{A}} = \mathbf{A}\mathbf{R}$. Then with probability $1 - \delta$,*

$$\|\mathbf{A}\mathbf{A}^T - \tilde{\mathbf{A}}\tilde{\mathbf{A}}^T\| \leq \mu\|\mathbf{A}\|^2.$$

1. *$\mathbf{R}$ is a dense random matrix with each entry being an i.i.d. sub-Gaussian random variable and $\ell = O(\frac{\mathrm{sr}(A) + \log(1/\delta)}{\epsilon^2})$.*

2. *$\mathbf{R}$ is a fully sparse embedding matrix , where each column has a single $\pm 1$ in a random position (sign and position chosen uniformly and independently) and $\ell = O(\frac{\mathrm{sr}(A)^2}{\epsilon^2 \delta})$. Additionally, the same matrix family except where the position and sign for each column are determined by a 4-independent hash function.*

3. *$\mathbf{R}$ is a Subsampled Randomized Hadamard Transform (SRHT) [41] with $\ell = O(\frac{\mathrm{sr}(A) + \log(1/(\epsilon\delta))}{\epsilon^2}) \log(\mathrm{sr}(A)/\delta)$.*

4. *$\mathbf{R}$ is a dense random matrix with each entry being $\pm\sqrt{\frac{1}{\ell}}$ for $\ell = O(\frac{\mathrm{sr}(A) + \log(1/\delta)}{\epsilon^2})$ and the entries are drawn from a $\log(\mathrm{sr}(A)/\delta)$-wise independent family of hash functions. Such a hash family can be constructed with $O(\log(d) \log(\mathrm{sr}(A)/\delta))$ random bits use standard techniques (see for e.g. Vadhan [28] Sec 3.5.5).*

Using Theorem 10 along with Lemma 8 and Lemma 9 and with the condition that $\mathrm{sr}(A) = O(k)$ gives Theorem 3 from the introduction—which shows that taking a random projection with $\ell = k^3 \cdot \mathrm{poly}(\Delta, \varepsilon^{-1})$ ensures that the error guarantees in Lemma 8 and Lemma 9 hold with high probability.

## C.3 Results on subsampling based sketches

Subsampling based sketches can yield both row and column space approximations. The algorithm for preserving the row space, i.e. approximating $\mathbf{A}^T\mathbf{A}$, using row subsampling is straightforward. The sketch samples $\ell$ rows of $\mathbf{A}$ proportional to their squared lengths to obtain a sketch $\tilde{\mathbf{A}} \in \mathbb{R}^{\ell \times d}$. This can be done in a single pass in a row streaming model using reservoir sampling. Our streaming algorithm for approximating anomaly scores using row subsampling follows the same outline as Algorithm 1 for Frequent Directions. We also obtain the same guarantees as Theorem 1 for Frequent Directions, using the guarantee for subsampling sketches stated at the end of the section (in Theorem 11). The guarantees in Theorem 1 are satisfied by subsampling $\ell = k^2 \cdot \mathrm{poly}(\kappa, \varepsilon^{-1}, \Delta)$ columns, and the algorithm needs $O(d\ell)$ space and $O(nd)$ time.

In order to preserve the column space using subsampling, i.e. approximate $\mathbf{A}\mathbf{A}^T$, we need to subsample the columns of the matrix. Our streaming algorithm follows a similar outline as Algorithm 2

which does a random projections of the rows and also approximates $\mathbf{A}\mathbf{A}^T$. However, there is a subtlety involved. We need to subsample the columns, but the matrix is arrives in row-streaming order. We show that using an additional pass, we can subsample the columns of $\mathbf{A}$ based on their squared lengths. This additional pass does reservoir sampling on the squared entries of the matrix, and uses the column index of the sampled entry as the column to be subsampled. The algorithm is stated in Algorithm 3. It requires space $O(\ell \log d)$ in order to store the $\ell$ indices to subsample, and space $O(\ell^2)$ to store the covariance matrix of the subsampled data. Using the guarantees for subsampling in Theorem 11, we can get the same guarantees for approximating anomaly scores as for a random projection of the rows. The guarantees for random projection in Theorem 10 are satisfied by subsampling $\ell = k^3 \cdot \text{poly}(\varepsilon^{-1}, \Delta)$ columns, and the algorithm needs $O(d\ell + \log d)$ space and $O(nd)$ time.

---

**Algorithm 3:** Algorithm to approximate anomaly scores using column subsampling

**Input**: Choice of $k$ and $\ell$.
**Initialization**
    Set covariance $\tilde{\mathbf{A}}^T \tilde{\mathbf{A}} \leftarrow 0$
    For $1 \leq t \leq \ell$, set $S_t = 1$       $\triangleright$ $S_i$ stores the $\ell$ column indices we will subsample
    Set $s \to 0$       $\triangleright$ $s$ stores the sum of the squares of entries seen so far
**Zeroth Pass:** *As each element $a_{ij}$ of $\mathbf{A}$ streams in,*
    Update $s \to s + a_{ij}^2$
    **for** $1 \leq t \leq \ell$ **do**
        Set $S_t \to j$ with probability $a_{ij}^2/s$
**First Pass:** *As each row $a_{(i)}$ streams in,*
    Project by $\mathbf{R}$ to get $\mathbf{R}^T a_{(i)}$
    Update covariance $\tilde{\mathbf{A}}^T \tilde{\mathbf{A}} \leftarrow \tilde{\mathbf{A}}^T \tilde{\mathbf{A}} + (\mathbf{R}^T a_{(i)})(\mathbf{R}^T a_{(i)})^T$
**SVD:**
    Compute the top $k$ right singular vectors of $\tilde{\mathbf{A}}^T \tilde{\mathbf{A}}$
**Second Pass:** *As each row $a_{(i)}$ streams in,*
    Project by $\mathbf{R}$ to get $\mathbf{R}^T a_{(i)}$
    For each projected row, use the estimated right singular vectors to compute the leverage scores and projection distances

---

**Guarantees for subsampling based sketches.** Drineas et al. [19] showed that sampling columns proportional to their squared lengths approximates $\mathbf{A}\mathbf{A}^T$ with high probability. They show a stronger Frobenius norm guarantee than the operator norm guarantee that we need, but this worsens the dependence on the stable rank. We will instead use the following guarantee due to Magen and Zouzias [31].

**Theorem 11.** *[31] Consider any matrix $\mathbf{A} \in \mathbb{R}^{n \times d}$. Let $\tilde{\mathbf{A}} \in \mathbb{R}^{n \times \ell}$ be a matrix obtained by subsampling the columns of $\mathbf{A}$ with probability proportional to their squared lengths. Then with probability $1 - 1/\text{poly}(\text{sr}(\mathbf{A}))$, for $\ell \geq \text{sr}(\mathbf{A}) \log(\text{sr}(\mathbf{A})/\mu^2)/\mu^2$*

$$\|\mathbf{A}\mathbf{A}^T - \tilde{\mathbf{A}}\tilde{\mathbf{A}}^T\| \leq \mu \|\mathbf{A}\|^2.$$

# D   Streaming Lower Bounds

In this section we prove lower bounds on streaming algorithms for computing leverage scores, rank $k$ leverage scores and ridge leverage scores for small values of $\lambda$. Our lower bounds are based on reductions from the multi party set disjointness problem denoted as $\text{DISJ}_{t,d}$. In this problem, each of $t$ parties is given a set from the universe $[d] = \{1, 2, \ldots, d\}$, together with the promise that either the sets are *uniquely intersecting*, i.e. all sets have exactly one element in common, or the sets are pairwise disjoint. The parties also have access to a common source of random bits. Chakrabarti et al. [43] showed a $\Omega(d/(t \log t))$ lower bound on the communication complexity of this problem. As usual, the lower bound on the communication in the set-disjointness problem translates to a lower bound on the space complexity of the streaming algorithm.

**Theorem 12.** *For sufficiently large $d$ and $n \geq O(d)$, let the input matrix be $\mathbf{A} \in \mathbb{R}^{n \times d}$. Consider a row-wise streaming model the algorithm may make a constant number passes over the data.*

1. *Any randomized algorithm which computes a $\sqrt{t}$-approximation to all the leverage scores for every matrix $\mathbf{A}$ with probability at least $2/3$ and with $p$ passes over the data uses space $\Omega(d/(t^2 p \log t))$.*

2. *For $\lambda \leq \frac{\text{sr}(\mathbf{A})}{2d} \sigma_1(\mathbf{A})^2$, any randomized streaming algorithm which computes a $\sqrt{t/2}$-approximation to all the $\lambda$-ridge leverage scores for every matrix $\mathbf{A}$ with $p$ passes over the data with probability at least $2/3$ uses space $\Omega(d/(pt^2 \log t))$.*

3. *For $2 \leq k \leq d/2$, any randomized streaming algorithm which computes any multiplicative approximation to all the rank $k$ leverage scores for every matrix $\mathbf{A}$ using $p$ passes and with probability at least $2/3$ uses space $\Omega(d/p)$.*

4. *For $2 \leq k \leq d/2$, any randomized algorithm which computes any multiplicative approximation to the distances from the principal $k$-dimensional subspace of every row for every matrix $\mathbf{A}$ with $p$ passes and with probability at least $2/3$ uses space $\Omega(d/p)$.*

We make a few remarks:

- The lower bounds are independent of the stable rank of the matrix. Indeed they hold both when $\text{sr}(\mathbf{A}) = o(d)$ and when $\text{sr}(\mathbf{A}) = \Theta(d)$.
- The Theorem is concerned only with the working space; the algorithms are permitted to have separate access to a random string.
- In the first two cases an additional $\log t$ factor in the space requirement can be obtained if we limit the streaming algorithm to one pass over the data.

Note that Theorem 12 shows that the Frequent Directions sketch for computing outlier scores is close to optimal as it uses $O(d\ell)$ space, where the projection dimension $\ell$ is a constant for many relevant parameter regimes. The lower bound also shows that the average case guarantees for the random projection based sketch which uses working space $O(\ell^2)$ cannot be improved to a point-wise approximation. The proof of the theorem is in Appendix G.

## E  Proofs of Section 2

We will prove bounds for items (2) and (3), which are given by Theorems 4 and 6 respectively. To prove these, the next two technical lemmas give perturbation bounds on the operator norm of positive semi-definite matrices of the from $\mathbf{U}_k \mathbf{\Lambda} \mathbf{U}_k^T$, where $\mathbf{\Lambda}$ is a diagonal matrix with non-negative entries. In order to do this, we split the matrix $\mathbf{\Sigma}$ to a well-separated component and a small residual component.

We now describe the decomposition of $\mathbf{\Sigma}$. Let $\delta$ be a parameter so that

$$6\mu \leq \delta \leq \Delta \tag{34}$$

We partition the indices $[k]$ into a set of disjoint intervals $B(\mathbf{A}, \delta) = \{B_1, \ldots, B_m\}$ based on the singular values of $\mathbf{A}$ so that there is a separation of at least $\delta\sigma_1^2$ between intervals, and at most $\delta\sigma_1^2$ within an interval. Formally, we start with $i = 1$ assigned to $B_1$. For $i \geq 2$, assume that we have assigned $i - 1$ to $B_j$. If

$$\sigma_i^2(\mathbf{A}) - \sigma_{i-1}^2(\mathbf{A}) \leq \delta\sigma_1^2(\mathbf{A})$$

then $i$ is also assigned to $B_j$, whereas if

$$\sigma_i^2(\mathbf{A}) - \sigma_{i-1}^2(\mathbf{A}) > \delta\sigma_1^2(\mathbf{A})$$

then it is assigned to a new bucket $B_{j+1}$. Let $b_j$ denote the largest index in the interval $B_j$ for $j \in [m]$.

Let $\mathbf{\Lambda} = \text{diag}(\lambda_1, \ldots \lambda_k)$ be a diagonal matrix with all non-negative entries which is constant on each interval $B_j$ and non-increasing across intervals. In other words, if $i \geq j$, then $\lambda_i \geq \lambda_j$, with equality holding whenever $i, j$ belong to the same interval $B_k$. Call such a matrix a *diagonal non-decreasing matrix with respect to $B(\mathbf{A}, \delta)$*. Similarly, we define diagonal non-increasing matrices with respect to $B(\mathbf{A}, \delta)$ to be non-increasing but are constant on each interval $B_k$. The following is the main technical lemma in this section, and handles the case where the diagonal matrix is well-separated.

It is a generalization of Lemma 6. In Lemma 6 we assumed that the eigenvalues in each bucket are equal, here we generalize to the case where the eigenvalues in each bucket are separated by at most $\delta\sigma_1^2(\mathbf{A})$.

**Lemma 10.** *Let $6\mu \le \delta \le \Delta$. Let $\mathbf{\Lambda} = \mathrm{diag}(\lambda_1, \ldots \lambda_k)$ be a diagonal non-increasing or a diagonal non-decreasing matrix with respect to $B(\mathbf{A}, \delta)$. Then*

$$\|\mathbf{U}_k\mathbf{\Lambda}\mathbf{U}_k^T - \tilde{\mathbf{U}}_k\mathbf{\Lambda}\tilde{\mathbf{U}}_k^T\| \le 4\|\mathbf{\Lambda}\|\sqrt{\frac{\mu}{\delta}}.$$

**Proof.** Let us set $\lambda_{b_{m+1}} = 0$ for convenience. Since

$$\mathbf{U}_{b_j}\mathbf{U}_{b_j}^T = \sum_{i=1}^{b_j} u^{(i)}u^{(i)^T}$$

we can write

$$\mathbf{U}_k\mathbf{\Lambda}\mathbf{U}_k^T = \sum_{j=1}^{m}\lambda_{b_j}\sum_{i\in B_j} u^{(i)}u^{(i)^T} = \sum_{j=1}^{m}(\lambda_{b_j} - \lambda_{b_{j+1}})\mathbf{U}_{b_j}\mathbf{U}_{b_j}^T$$

and similarly for $\tilde{\mathbf{U}}_k\mathbf{\Lambda}\tilde{\mathbf{U}}_k^T$. So we can write

$$\mathbf{U}_k\mathbf{\Lambda}\mathbf{U}_k^T - \tilde{\mathbf{U}}_k\mathbf{\Lambda}\tilde{\mathbf{U}}_k^T = \sum_{j=1}^{m}(\lambda_{b_j} - \lambda_{b_{j+1}})(\mathbf{U}_{b_j}\mathbf{U}_{b_j}^T - \tilde{\mathbf{U}}_{b_j}\tilde{\mathbf{U}}_{b_j}^T).$$

Therefore, by the triangle inequality and Lemma 5,

$$\|\mathbf{U}_k\mathbf{\Lambda}\mathbf{U}_k^T - \tilde{\mathbf{U}}_k\mathbf{\Lambda}\tilde{\mathbf{U}}_k^T\| = \sum_{j=1}^{m}|\lambda_{b_j} - \lambda_{b_{j+1}}|\|(\mathbf{U}_{b_j}\mathbf{U}_{b_j}^T - \tilde{\mathbf{U}}_{b_j}\tilde{\mathbf{U}}_{b_j}^T)\| \le 2\sqrt{\frac{\mu}{\delta}}\sum_{j=1}^{m}|\lambda_{b_j} - \lambda_{b_{j+1}}|$$

thus proving that $\sum_{j=1}^{m}|\lambda_{b_j} - \lambda_{b_{j+1}}| \le 2\|\mathbf{\Lambda}\|$ would imply the claim.

When $\mathbf{\Lambda}$ is diagonal non-increasing with respect to $B(\mathbf{A}, \delta)$, then $\lambda_{b_j} - \lambda_{b_{j+1}} \ge 0$ for all $j \in [m]$, and $\|\mathbf{\Lambda}\| = \lambda_{b_1}$. Hence

$$\sum_{j=1}^{m}|\lambda_{b_j} - \lambda_{b_{j+1}}| = \sum_{j=1}^{m}(\lambda_{b_j} - \lambda_{b_{j+1}}) = \lambda_{b_1} - \lambda_{b_{m+1}} = \|\mathbf{\Lambda}\|.$$

When $\mathbf{\Lambda}$ is diagonal non-decreasing with respect to $B(\mathbf{A}, \delta)$, then for $j \le m-1$, $\lambda_{b_j} \le \lambda_{b_{j+1}}$, and $\|\mathbf{\Lambda}\| = \lambda_{b_m}$. Hence

$$\sum_{j=1}^{m-1}|\lambda_{b_j} - \lambda_{b_{j+1}}| = \sum_{j=1}^{m-1}\lambda_{b_{j+1}} - \lambda_{b_j} = \lambda_{b_m} - \lambda_{b_1} \le \lambda_{b_m}$$

whereas $|\lambda_{b_m} - \lambda_{b_{m+1}}| = \lambda_{b_m}$. Thus overall,

$$\sum_{j=1}^{m}|\lambda_{b_j} - \lambda_{b_{j+1}}| \le 2\lambda_{b_m} = 2\|\mathbf{\Lambda}\|.$$

∎

Figure 5: Illustration of the decomposition of the $k$ singular values into $m$ intervals such that there is a separation of at least $\delta\sigma_1^2$ between intervals, and at most $\delta\sigma_1^2$ within an interval.

We use this to prove our perturbation bound for $\mathbf{U}_k \boldsymbol{\Sigma}_k^2 \mathbf{U}_k^T$.

**Theorem 4.** *Let* $\mu \leq \min(\Delta^3 k^2, 1/(20k))$.

$$\|\mathbf{U}_k \boldsymbol{\Sigma}_k^2 \mathbf{U}_k^T - \tilde{\mathbf{U}}_k \boldsymbol{\Sigma}_k^2 \tilde{\mathbf{U}}_k^T\| \leq 8\sigma_1^2 (\mu k)^{1/3}.$$

**Proof.** Define $\boldsymbol{\Lambda}$ to be the $k \times k$ diagonal non-increasing matrix such that all the entries in the interval $B_j$ are $\sigma_{b_j}^2$. Define $\boldsymbol{\Omega}$ to be the $k \times k$ diagonal matrix such that $\boldsymbol{\Lambda} + \boldsymbol{\Omega} = \boldsymbol{\Sigma}_k^2$. With this notation,

$$\|\mathbf{U}_k \boldsymbol{\Sigma}_k^2 \mathbf{U}_k^T - \tilde{\mathbf{U}}_k \boldsymbol{\Sigma}_k^2 \tilde{\mathbf{U}}_k^T\| = \|(\mathbf{U}_k \boldsymbol{\Lambda} \mathbf{U}_k^T - \tilde{\mathbf{U}}_k \boldsymbol{\Lambda} \tilde{\mathbf{U}}_k^T) + (\mathbf{U}_k \boldsymbol{\Omega} \mathbf{U}_k^T - \tilde{\mathbf{U}}_k \boldsymbol{\Omega} \tilde{\mathbf{U}}_k^T)\|$$

$$\leq \|\mathbf{U}_k \boldsymbol{\Lambda} \mathbf{U}_k^T - \tilde{\mathbf{U}}_k \boldsymbol{\Lambda} \tilde{\mathbf{U}}_k^T\| + \|\mathbf{U}_k \boldsymbol{\Omega} \mathbf{U}_k^T - \tilde{\mathbf{U}}_k \boldsymbol{\Omega} \tilde{\mathbf{U}}_k^T)\| \qquad (35)$$

By definition, $\boldsymbol{\Lambda}$ is diagonal non-increasing, $\|\boldsymbol{\Lambda}\| = \sigma_{b_1}^2 \leq \sigma_1^2$. Hence by part (1) of Lemma 10,

$$\|\mathbf{U}_k \boldsymbol{\Lambda} \mathbf{U}_k^T - \tilde{\mathbf{U}}_k \boldsymbol{\Lambda} \tilde{\mathbf{U}}_k^T\| \leq 4\sigma_1^2 \sqrt{\frac{\mu}{\delta}}.$$

By our definition of the $B_j$s, if $i, i+1 \in B_j$ then $\sigma_i^2 - \sigma_{i+1}^2 \leq \delta\sigma_1^2$, hence for any pair $i, i'$ $in B_j$, $\sigma_i^2 - \sigma_{i'}^2 \leq k\delta\sigma_1^2$. Hence

$$\|\boldsymbol{\Omega}\| = \max_{i \in B_j}(\sigma_i^2 - \sigma_{b_j}^2) \leq k\delta\sigma_1^2$$

We use Lemma 4 to get

$$\|\mathbf{U}_k \boldsymbol{\Omega} \mathbf{U}_k^T - \tilde{\mathbf{U}}_k \boldsymbol{\Omega} \tilde{\mathbf{U}}_k^T\| \leq 2k\delta\sigma_1^2.$$

Plugging these bounds into Equation (35), we get

$$\|\mathbf{U}_k \boldsymbol{\Sigma}_k^2 \mathbf{U}_k^T - \tilde{\mathbf{U}}_k \boldsymbol{\Sigma}_k^2 \tilde{\mathbf{U}}_k^T\| \leq 4\sigma_1^2 (\sqrt{\frac{\mu}{\delta}} + k\delta).$$

We choose $\delta = \mu^{1/3}/k^{2/3}$ that minimizes the RHS, to get

$$\|\mathbf{U}_k \boldsymbol{\Sigma}_k^2 \mathbf{U}_k^T - \tilde{\mathbf{U}}_k \boldsymbol{\Sigma}_k^2 \tilde{\mathbf{U}}_k^T\| \leq 8\sigma_1^2 (\mu k)^{1/3}.$$

We need to ensure that this choice satisfies $6\mu \leq \delta$. This holds since it is equivalent to $6(\mu k)^{2/3} \leq 1$, which is implied by $\mu k \leq 1/20$. We need $\delta \leq \Delta$ which holds since $\mu \leq \Delta^3 k^2$. ∎

Next we derive our perturbation bound for $\mathbf{U}_k \boldsymbol{\Sigma}_k^{-2} \mathbf{U}_k^T$, which will depend on the condition number $\kappa = \sigma_1^2/\sigma_k^2$. The proof is similar to the proof of Theorem 4.

**Theorem 6.** *Let* $\kappa_k$ *denote the condition number* $\kappa_k = \sigma_1^2/\sigma_k^2$. *Let* $\mu \leq \min(\Delta^3 (k\kappa_k)^2, 1/(20k\kappa_k))$. *Then,*

$$\|\mathbf{U}_k \boldsymbol{\Sigma}_k^{-2} \mathbf{U}_k^T - \tilde{\mathbf{U}}_k \boldsymbol{\Sigma}_k^{-2} \tilde{\mathbf{U}}_k^T\| \leq \frac{8}{\sigma_k^2} (\mu k \kappa_k)^{1/3}.$$

**Proof.** We use a similar decomposition as in Theorem 4. Define $\boldsymbol{\Lambda}$ to be diagonal non-decreasing such that all the entries in the interval $B_j$ are $1/\sigma_{b_j}^2$. Note that $\|\boldsymbol{\Lambda}\| \leq \sigma_k^{-2}$. Using Lemma 10, we get

$$\|\mathbf{U}_k \boldsymbol{\Lambda}_k^{-2} \mathbf{U}_k^T - \tilde{\mathbf{U}}_k \boldsymbol{\Lambda} \tilde{\mathbf{U}}_k^T\| \leq \frac{4}{\sigma_k^2} \sqrt{\frac{\mu}{\delta}}. \qquad (36)$$

Define $\boldsymbol{\Omega} = \boldsymbol{\Sigma}_k^{-2} - \boldsymbol{\Lambda}$. Note that

$$\|\boldsymbol{\Omega}\| = \max_{i \in B_j} \frac{1}{\sigma_{b_j}^2} - \frac{1}{\sigma_i^2} = \max_{i \in B_j} \frac{\sigma_i^2 - \sigma_{b_j}^2}{\sigma_i^2 \sigma_{b_j}^2} \leq \frac{k\delta\sigma_1^2}{\sigma_k^4} = \frac{k\kappa\delta}{\sigma_k^2}.$$

By using this in Lemma 4,

$$\|\mathbf{U}_k \boldsymbol{\Omega} \mathbf{U}_k^T - \tilde{\mathbf{U}}_k \boldsymbol{\Omega} \tilde{\mathbf{U}}_k^T\| \leq \frac{2k\kappa\delta}{\sigma_k^2}. \qquad (37)$$

Putting Equation (36) and (37) together, we get

$$\|\mathbf{U}_k \boldsymbol{\Sigma}_k^{-2} \mathbf{U}_k^T - \tilde{\mathbf{U}}_k \boldsymbol{\Sigma}_k^{-2} \tilde{\mathbf{U}}_k^T\| \leq \frac{4}{\sigma_k^2} \left( \sqrt{\frac{\mu}{\delta}} + k\kappa\delta \right).$$

The optimum value of $\delta$ is $\mu^{1/3}/(k\kappa)^{2/3}$ which gives the claimed bound. A routine calculation shows that the condition $6\mu \leq \delta \leq \Delta$ holds because $\mu \leq \min(\Delta^3 (k\kappa)^2, 1/(20k\kappa))$. ∎

# F  Ridge Leverage Scores

Regularizing the spectrum (or alternately, assuming that the data itself has some ambient Gaussian noise) is closely tied to the notion of ridge leverage scores [44]. Various versions of ridge leverages had been shown to be good estimators for the Mahalanobis distance in the high dimensional case and were demonstrated to be an effective tool for anomaly detection [25]. There are efficient sketches that approximate the ridge leverage score for specific values of the parameter $\lambda$ [45].

Recall that we measured deviation in the tail by the distance from the principal $k$-dimensional subspace, given by

$$T^k(i) = \sum_{j=k+1}^{d} \alpha_j^2.$$

We prefer this to using

$$L_i^{>k} := \sum_{j=k+1}^{d} \frac{\alpha_j^2}{\sigma_j}^2$$

since it is more robust to the small $\sigma_j$, and is easier to compute in the streaming setting.[5]

An alternative approach is to consider the ridge leverage scores, which effectively replaces the covariance matrix $\mathbf{A}^T\mathbf{A}$ with $\mathbf{A}^T\mathbf{A} + \lambda\mathbf{I}$, which increases all the singular values by $\lambda$, with the effect of damping the effect of small singular values. We have

$$L_\lambda(i) = \sum_{j=1}^{d} \frac{\alpha_j^2}{\sigma_j^2 + \lambda}.$$

Consider the case when the data is generated from a true $k$-dimensional distribution, and then corrupted with a small amount of white noise. It is easy to see that the data points will satisfy both concentration and separation assumptions. In this case, all the notions suggested above will essentially converge. In this case, we expect $\sigma_{k+1}^2 \approx \cdots \approx \sigma_n^2$. So

$$L_i^{>k} = \sum_{j=k+1}^{d} \frac{\alpha_j^2}{\sigma_j^2} \approx \frac{T^k(i)}{\sigma_{k+1}^2}.$$

If $\lambda$ is chosen so that $\sigma_k^2 \gg \lambda \gg \sigma_{k+1}^2$, it follows that

$$L_\lambda(i) \approx L^k(i) + \frac{T^k(i)}{\lambda}.$$

# G  Proof of Theorem 12: Streaming Lower Bounds

**Proof.** We describe the reduction from $\mathrm{DISJ}_{t,d}$ to computing each of the four quantities.

(1) **Leverage scores:** Say for contradiction we have an algorithm which computes a $\sqrt{t}$-approximation to all the leverage scores for every matrix $\mathbf{A} \in \mathbb{R}^{n \times d}$ using space $O(d/(kt^2 \log t))$ and $k = O(1)$ passes. We will use this algorithm to design a protocol for $\mathrm{DISJ}_{t,d}$ using a communication complexity of $O(d/(t \log t))$. In other words we need the following lemma.

**Lemma 11.** *A streaming algorithm which approximates all leverage scores within $\sqrt{t}$ with $p$ passes over the data, and which uses space $s$ implies a protocol for $\mathrm{DISJ}_{t,d}$ with communication complexity $s \cdot p \cdot t$*

**Proof.** Given an $\mathrm{DISJ}_{t,d}$ instance, we create a matrix $\mathbf{A}$ with $d$ columns as follows: Let $e_i$ be the $i$th row of the $(d \times d)$ identity matrix $\mathbf{I}_{d \times d}$. The vector $e_i$ is associated with the $i$th element of the universe $[d]$. Each player $j$ prepares a matrix $\mathbf{A}_j$ with $d$ columns by adding the row $e_i$ for each $i \in [d]$ in its set. $\mathbf{A}$ is composed of the rows of the $t$ matrices $\mathbf{A}_j$.

We claim that $\sqrt{t}$ approximation to the leverage scores of $\mathbf{A}$ suffices to differentiate between the case the sets are disjoint and the case they are uniquely intersecting. To see this note that if the sets are all disjoint then each row is linearly independent from the rest and therefore all rows have leverage score $1$. If the sets are *uniquely intersecting*, then exactly one row in $\mathbf{A}$ is repeated $t$ times. A moment's reflection reveals that in this case, each of these rows has leverage score $1/t$. Hence a $\sqrt{t}$-approximation to all leverage scores allows the parties to distinguish between the two cases.

The actual communication protocol is now straight forward. Each party $i$ in turn runs the algorithm over its own matrix $\mathbf{A}_i$ and passes the $s$ bits which are the state of the algorithm to the party $i+1$. The last party outputs the result. If the algorithm requires $p$ passes over the data the total communication is $p \cdot s \cdot t$. ∎

Theorem 12 now follows directly from the lower bound on the communication complexity of $\mathrm{DISJ}_{t,d}$.

Note that in the construction above the stable rank $\mathrm{sr}(\mathbf{A})$ is $\Theta(d)$. The dependency on $d$ could be avoided by adding a row and column to $\mathbf{A}$: A column of all zeros is added to $\mathbf{A}$ and then the last party adds a row to $\mathbf{A}_t$ having the entry $\sqrt{K} \geq 1$ in the last column. Now since $K > 1$ the last row will dominate both the Frobenius and the operator norm of the matrix but does not affect the leverage score of the other rows. Note that $\mathrm{sr}(\mathbf{A}) \leq \frac{K+d}{K}$. By choosing $K$ large enough, we can now decrease $\mathrm{sr}(\mathbf{A})$ to be arbitrarily close to 1. Note also that if the algorithm is restricted to one pass, the resulting protocol is one directional and has a slightly higher lower bound of $\Omega(d/t^2)$.

(2) **Ridge leverage scores:** We use the same construction as before and multiply $\mathbf{A}$ by $\sigma \geq \sqrt{\lambda}$. Note that as required, the matrix $\mathbf{A}$ has operator norm $\sigma$. As before, it is sufficient to claim that by approximately computing all ridge leverage scores the parties can distinguish between the case their sets are mutually disjoint and the case they are uniquely intersecting. Indeed, if the sets are mutually disjoint then all rows have ridge leverage scores $\frac{\sigma^2}{\sigma^2+\lambda}$. If the sets are uniquely intersecting, then exactly one element is repeated $t$ times in the matrix $\mathbf{A}$, in which case this element has ridge leverage score $\frac{\sigma^2}{t\sigma^2+\lambda}$. These two cases can be distinguished by a $\sqrt{t/2}$-approximation to the ridge leverage scores if $\lambda \leq \sigma^2$.

To modify the stable rank $\mathrm{sr}(\mathbf{A})$ in this case we do the same trick as before, add a column of zeros and the last party adds an additional row having the entry $\sqrt{K}\sigma \geq \sigma$ in its last column. Note that $\mathrm{sr}(\mathbf{A}) \leq \frac{K+d}{K}$, and by increasing $K$ we can decrease $\mathrm{sr}(\mathbf{A})$ as necessary. However, $\|\mathbf{A}\|^2$ now equals $K\sigma^2$, and hence we need to upper bound $K$ in terms of the stable rank $\mathrm{sr}(\mathbf{A})$ to state the final bound for $\lambda$ in terms of $\|\mathbf{A}\|^2$. Note that $\mathrm{sr}(\mathbf{A}) \leq \frac{K+d}{K} \implies K \leq 2d/\mathrm{sr}(\mathbf{A})$. Hence $\lambda \leq \frac{\mathrm{sr}(\mathbf{A})}{2d}\|\mathbf{A}\|^2$ ensures that $\lambda \leq \sigma^2$.

(3) **Rank-$k$ leverage scores:** The construction is similar to the previous ones, with some modifications to ensure that the top $k$ singular vectors are well defined. We set number of parties to be 2, and let the universe be of size $d' = d - k$, so the matrix is wider that the size of universe by $k$ columns. As before, for $i \leq d'$ the $i$'th row of $\mathbf{I}_{d\times d}$ is associated with the $i$'th element of the universe $[d']$. The first set of rows in $\mathbf{A}$ are the rows corresponding to the elements in the first party's set and the next set of rows in $\mathbf{A}$ correspond to the elements in the second party's set. The second party also adds the last $k$ rows of $\mathbf{I}_{d\times d}$, scaled by $1.1$, to the matrix $\mathbf{A}$.

We claim that by computing a multiplicative approximation to all rank $k$ leverage scores the parties can determine whether their sets are disjoint. If the sets are all disjoint, then the top $k$ right singular vectors correspond to the last $k$ rows of the matrix $\mathbf{I}_{d\times d}$, and these are orthogonal to the rest of the matrix and hence the rank $k$ leverage scores of all rows except the additional ones added by the second party are all $0$. If the sets are intersecting, then the row corresponding to the intersecting element is the top right singular vector of $\mathbf{A}$, as it has singular value $\sqrt{2} > 1.1$. Hence the rank $k$ leverage score of this row is $1/2$. Hence the parties can distinguish whether they have disjoint sets by finding any multiplicative approximation to all rank $k$ leverage scores.

We apply a final modification to decrease the stable rank $\mathrm{sr}(\mathbf{A})$ as necessary. We scale the $d$th row of $\mathbf{I}_{d\times d}$ by a constant $\sqrt{K}$. Note that $\mathrm{sr}(\mathbf{A}) \leq \frac{K+d+2k}{K}$. By choosing $K$ accordingly, we can now decrease $\mathrm{sr}(\mathbf{A})$ as desired. We now examine how this scaling affects the rank $k$ leverage scores for the rows corresponding to the sets. When the sets are not intersecting, the rank $k$ leverage scores

of all the rows corresponding to the set elements are still 0. When the sets are intersecting, the row corresponding to the intersecting element is at the least second largest right singular vector of $\mathbf{A}$ even after the scaling, as $\sqrt{2} > 1.1$. In this case, for $k \geq 2$ the rank $k$ leverage score of this row is 1/2, hence the parties can distinguish whether they have disjoint sets by finding any multiplicative approximation to all rank $k$ leverage scores for any $2 \leq k \leq d/2$.

**Distance from principal $k$-dimensional subspace:** We use the same construction as in statement (3). If the sets are non-intersecting, all the rows corresponding to the sets of the two-parties have distance 1 from the principal $k$-dimensional subspace. If the sets are intersecting, the row corresponding to the element in the intersection has distance 0 from the principal $k$-dimensional subspace, as that row is either the top or the second right singular vector of $\mathbf{A}$. Hence, any multiplicative approximation could be used to distinguish between the two cases.

∎