[Reviews · NeurIPS 2018]

Reviewer 1



Given an n x d matrix with n points in d dimensions (n > d), a popular outlier detection technique to find outlier points requires O(nd^2) time and O(d^2) space. This technique is based on the so called leverage scores. When d is large, outlier detection via leverage scores becomes hard. This paper provides algorithm that reduce the space/time complexity of leverage score computation. I found the results interesting and novel. I would like to see a better comparison with other "fast" leverage scores approximation algorithms, for example https://www.cs.purdue.edu/homes/pdrineas/documents/publications/Drineas_JMLR_2012.pdf

Reviewer 2



Authors propose to use matrix sketching methods to approximate some subspace based scores for outlier detection. They suggest " projection distance" and "leverage scores" as outlier scores. Neither the idea, nor the methods are original or new. leverage scores and projection distances have been used before for outlier detection, and authors use existing methods in literature to approximate these scores. It seem novelty of the work is in proofs. I did not double check correctness of all proofs. I have a question for the authors : in the experiment section, they mention to measure accuracy, they choose a value of eta' which maximizes F1 score. At the end they are reporting F1 scores of the competing methods. So why are they changing the threshold eta' to obtain the maximum F1. I dont see a training and testing phase, that you'll say for example eta' is found in the training phase and then it is used in testing to measure F1 score.... Please clarify.

Reviewer 3



Authors develop two new algorithms to find anomalies in high dimensional data when the row data is streaming in. Those algorithms are both two-pass algorithms to approximately compute the rank k leverage score as well as rank k projection distance with running complexity time of O(nd\ell), where n and d are the size of an input matrix A and \ell is independent of n and d, depends only on rank k and some other parameters. While the first algorithm requires O(d\ell) memory, the second one uses only O(log(d)) memory. However, the cost of achieving such reduction is losing accuracy as the first algorithm posses pointwise guarantees from row space approximation and the second one holds average-case guarantees from columns space approximation. The approximation rates are derived analytically and illustrated numerically. In general, the paper is well-written, well-referenced throughout, and in appropriate format/structure. The practical algorithms presented in this paper contributes to the existing literature by proposing two simple methods which do work for any sketch matrix satisfying certain inequalities ((1) and (2)). The paper can be considered for publication in NIPS 2018 Proceedings if the authors can address the following minor comments in the manuscript: 1. Line 145: Replace "other parameters" with "k and \epsilon". 2. Line 184: Replace "2" with "(2)". 3. Line 187: Remove "for" before "e.g.". 4. Line 189: Replace "a n\times n" with "an n\times n". 5. Line 191: Replace "a \ell\times \ell" with "an \ell\times \ell". 6. Line 198: There is a typo in "psedorandomness". 7. Line 230: To keep the statement of Algorithm 1 consistent with the numerical results, replace "Frequent Directions" in Input of this algorithm with "Random Column Projections". Accordingly, lines 156-157 as well as Footnote 2 should be modified. 8. Figures 2-4: In all six plots, replace "Column Projection" and "Row Projection" with "Algorithm 1" and "Algorithm 2", respectively. At the end end, I have a minor question: To avoid degeneracy in the spectrum of matrix A in theorems 1 and 3, should we assume that that A is (k,\delta) and (k-1,\delta)-separated?